



# Light absorption by marine cyanobacteria affects tropical climate mean state and variability

Hanna Paulsen[1], Tatiana Ilyina[1], Johann H. Jungclaus[1], Katharina D. Six[1], and Irene Stemmler[1]

[1]Max Planck Institute for Meteorology, Hamburg, Germany

**Correspondence:** Hanna Paulsen (hanna.paulsen@mpimet.mpg.de)

**Abstract.**

Observations indicate that positively buoyant marine cyanobacteria, which are abundant throughout the tropical and subtropical ocean, have a strong local heating effect due to light absorption at the ocean surface. How these local changes in radiative heating affect the climate system on the large scale is unclear as of yet. We use the Max Planck Institute Earth System Model (MPI-ESM) and find that – in contrast to the heating effect which was reported in previous studies – cyanobacteria have a considerable cooling effect on tropical climatological sea surface temperature (SST) in the order of 0.5 K. This cooling is caused by local shading of subtropical subsurface water that is upwelled at the equator and in eastern boundary upwelling systems. Implications for the climate system include an expansion of the Hadley cells and a westward shift of the Walker circulation. The amplitude of the seasonal cycle of SST is increased in large parts of the tropical ocean by up to 25 %, and the tropical Pacific interannual variability is enhanced by ∼20 %. This study emphasizes the sensitivity of the tropical climate system to light absorption by the specific phytoplankton group of cyanobacteria due to its regulative effect on tropical SST. Generally, including the phytoplankton-dependent light attenuation instead of a globally uniform attenuation depth improves some of the major model temperature biases, indicating the relevance of taking into account this bio-physical feedback in climate models.

## 1 Introduction

Phytoplankton pigments, predominantly chlorophyll *a* (Chl), absorb light and thereby modify the vertical distribution of radiative heating in the upper ocean (e.g., Lewis et al., 1990; Strutton and Chavez, 2004). Numerous model studies, using ocean-only and coupled climate models, indicate that the biologically-induced redistribution of heat in the water column considerably affects ocean temperature and circulation (e.g., Murtugudde et al., 2002; Manizza et al., 2008; Löptien et al., 2009) with implications for climate mean state and variability (e.g., Wetzel et al., 2006; Anderson et al., 2007; Patara et al., 2012). In this study, we specifically investigate the effects of light absorption by the phytoplankton group of positively buoyant marine cyanobacteria on the tropical climate system.

Positively buoyant marine cyanobacteria are observed by in situ and satellite measurements to have a particularly strong local heating effect on sea surface temperature (SST) by up to 0.95–5.0 K (Kahru et al., 1993; Capone et al., 1998; Wurl et al., 2018). Cyanobacteria are abundant throughout the tropical and subtropical ocean, where they often represent a dominant fraction of total phytoplankton biomass (e.g., Carpenter and Romans, 1991; Capone et al., 1997; Luo et al., 2012). The unique capability



of cyanobacteria to fix nitrogen gas ($N_2$) enables them to grow within and at the edges of the nitrate-depleted subtropical gyres (e.g., Luo et al., 2012). Positively buoyant cyanobacteria have the capacity to form large surface blooms extending over up to several millions of square kilometres (e.g., Capone et al., 1998). Although the chlorophyll content of cyanobacteria is in general rather low compared to that of other phytoplankton (e.g., Berman-Frank et al., 2001; Carpenter et al., 2004; Sathyendranath

et al., 2009), the dense accumulations of biomass in cyanobacteria blooms result in high chlorophyll concentrations (e.g., Subramaniam et al., 2001; Westberry and Siegel, 2006) and hence strong light absorption and heat trapping at the ocean surface.

The strong observed effect of cyanobacteria on the local vertical thermal structure of the water column is supported by one-dimensional and regional model studies of the Baltic Sea and the North Atlantic Ocean (Hense, 2007; Sonntag and Hense,

2011; Sonntag, 2013). In these studies, positively buoyant cyanobacteria lead to a local heating effect of up to 2 K and a mixed layer depth shoaling of up to 20 m. These changes in the physical environment regionally promote growth of cyanobacteria itself, constituting a positive feedback loop between ocean biology and physics.

The existing observational and model studies give indeed evidence for the importance of light absorption by cyanoabacteria. But these studies are representative only on a local, at the best, regional scale. All global model studies, however, only

consider the effect of light absorption by phytoplankton in general, without differentiating between individual phytoplankton groups (e.g., Patara et al., 2012, and references therein). Moreover, none of the biogeochemical models used in these studies contains positively buoyant cyanobacteria. Cyanobacteria are, however, abundant throughout large parts of the tropical and subtropical ocean. In these areas, more precisely within and at the margins of the subtropical gyres, light absorption by marine biota was proposed to be particularly relevant in affecting tropical SST and the climate system (Anderson et al., 2007, 2009;

Gnanadesikan and Anderson, 2009). By using a modified satellite chlorophyll climatology in which chlorophyll is set to zero in distinct regions, these studies showed that the presence of chlorophyll in the respective regions strongly affects the oceanic and atmospheric large scale circulation as well as tropical Pacific interannual variability. Given that cyanobacteria are a dominant phytoplankton group in these identified to be relevant regions of the ocean, the open question is: Do cyanobacteria affect their environment not only locally, as indicated by previous studies, but play a role in the climate system on a more globale scale?

We therefore investigate the interactive feedback from cyanobacteria light absorption on the large scale tropical climate system. We use the Max Planck Institute Earth System Model (MPI-ESM) which was recently extended by a realistic representation of positively buoyant, $N_2$-fixing cyanobacteria (Paulsen et al., 2017). We include cyanobacteria in addition to bulk phytoplankton in affecting the shortwave attenuation depth. By setting the chlorophyll content of cyanobacteria to zero, we separate the effects induced by cyanobacteria from those induced by bulk phytoplankton and address following questions. How

does the local redistribution of heat by cyanobacteria affect the large scale ocean temperature distribution and the climate mean state, such as ocean and atmosphere general circulation? What are the effects on climate variability, such as on seasonal variability and on tropical Pacific interannual variability? How do the changes in ocean temperature and circulation feed back on phytoplankton growth itself? Are there positive/negative feedbacks at play? By varying the chlorophyll content per cyanobacteria biomass (and thereby the strength of light absorption), we get a better understanding of the underlying processes and can

assess the sensitivity of the results to this parameter. We furthermore compare the model results against a model state with



phytoplankton-independent (globally uniform) light attenuation. We discuss the results in the context of model biases and in the context of previous model studies, and indicate implications for the Earth system model.

## 2 Model description and experimental setup

### 2.1 The MPI-ESM

We use MPI-ESM1.2, which consists of the coupled general circulation models for the atmosphere ECHAM (Stevens et al., 2013) and the ocean MPIOM (Jungclaus et al., 2013), the ocean biogeochemistry model HAMOCC (Ilyina et al., 2013) extended by prognostic $N_2$-fixing cyanobacteria (Paulsen et al., 2017), and the land surface and terrestrial biosphere model including dynamic vegetation JSBACH (Reick et al., 2013; Schneck et al., 2013). The explicit model versions are ECHAM6.3.02p2, MPIOM1.6.2p2, HAMOCC5.2, JSBACH3.10, which are further developments of the models described in the references above.

We apply a grid configuration referred to as LR (GR15 for MPIOM, T63L47 for ECHAM6). In the atmosphere, the horizontal resolution is T63 in spectral space (approximately 1.75° on a Gaussian grid) with 47 vertical $\sigma$-hybrid layers. The time step is 450 seconds. In the ocean, the bipolar grid GR15 has poles over Antarctica and Greenland and a horizontal resolution of approximately 1.5 °, gradually varying between 15 km in the Arctic and about 184 km in the Tropics. In the vertical, there are 40 unevenly spaced layers with level thicknesses increasing with depth and nine layers located within the upper 90 m. The

time step is 2700 seconds. MPIOM is a z-coordinate global general circulation model solving the primitive equations under the hydrostatic and Boussinesq approximation on a C-grid with a free surface (Marsland et al., 2003; Jungclaus et al., 2013). Momentum, heat and freshwater fluxes are coupled daily between ECHAM and MPIOM using the Ocean Atmosphere Sea Ice Soil (OASIS3-MCT; Valcke, 2013) coupler. Incoming shortwave radiation is passed daily from ECHAM to MPIOM.

  The global ocean biogeochemistry model HAMOCC serves to simulate carbon cycling in the ocean. The spatial and temporal

resolutions of HAMOCC are inherited from MPIOM. HAMOCC includes biogeochemical processes in the water column, the sediment, and gas-exchange processes at the air-sea interface. Biogeochemical tracers in the water column are fully advected, mixed, and diffused by the flow field of the physical model. Biogeochemistry dynamics, which are premised on an extended NPZD (Nutrients, Phytoplankton, Zooplankton, Detritus) model approach (Six and Maier-Reimer, 1996), include the compartments nutrients (phosphate, nitrate, and iron), oxygen, silicate, opal, calcium carbonate, dissolved inorganic carbon, alkalinity,

phytoplankton (bulk phytoplankton and $N_2$-fixing cyanobacteria), zooplankton, dissolved organic matter, and detritus. Organic material composition follows a constant molar ratio (C:N:P:$O_2$ = 122:16:1:-172) based on Takahashi et al. (1985) and of iron (Fe:P = 366·$10^{-6}$:1) based on Johnson et al. (1997).

  Phytoplankton is represented by two tracers in the model, $N_2$-fixing cyanobacteria (Cya) and bulk phytoplankton (Phy). The growth parameterization of cyanobacteria is based on physiological characteristics of *Trichodesmium* (Paulsen et al.,

2017), a positively buoyant cyanobacteria group which is considered as one of the most important marine diazotrophs (e.g., LaRoche and Breitbarth, 2005). One major difference between cyanobacteria as compared to bulk phytoplankton is their ability to fix $N_2$. When available, cyanobacteria take up nitrate and, on the other hand, fix $N_2$ under nitrate depletion. They are hence not limited by nitrate. Furthermore, cyanobacteria are positively buoyant with a rising velocity of 1 m d$^{-1}$ (based

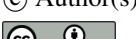



on Sonntag, 2013), in contrast to bulk phytoplankton which are neutrally buoyant. Growth of cyanobacteria occurs within a specific optimum temperature range described by a modified Gaussian function with an optimum at 28°C (Sonntag and Hense, 2011; Breitbarth et al., 2007). For bulk phytoplankton, on the contrary, a power law temperature dependency is applied (Eppley, 1972). Cyanobacteria grow slower than bulk phytoplankton, have a higher iron limitation, and are assumed not to be grazed. A detailed description of the parameterization as well as evaluation of the model performance with respect to cyanobacteria are given in Paulsen et al. (2017).

## 2.2 Parameterization of radiative heating in the water column

Incoming shortwave radiation in the open ocean is attenuated primarily by seawater and by phytoplankton. We use chlorophyll, derived from the phytoplankton concentrations simulated in the biogeochemical model, as a measure of strength of light absorption in the physical model. The vertical light field I(z) is described by the scheme after Zielinski et al. (2002):

$$I(z) = I_0 \cdot f_{vis} \cdot \left[ \sigma \cdot exp(-z \cdot k_r) + (1 - \sigma) \cdot exp\left(-z \cdot k_w - k_{Chl} \cdot \int_0^z Chl(z)dz\right) \right]. \qquad (1)$$

Here, $I_0$ is the incoming shortwave radiation that reaches the sea surface. $f_{vis}$ is the visible light fraction (0.58) which has the potential to penetrate into deeper layers. It covers the wavelength range of 400–700 nm. At 580 nm, the light spectrum is divided (prescribed by $\sigma = 0.4$) into two domains: For larger wavelengths (red domain) attenuation is dominated by sea water with the attenuation coefficient $k_r$ (0.35 m$^{-1}$). For shorter wavelengths (blue/green domain) the absorption by chlorophyll with the absorption coefficient $k_{Chl}$ (0.04 m$^{-1}$) is considered in addition to clear water with the absorption coefficient $k_w$ (0.03 m$^{-1}$). Chlorophyll (Chl) is assumed to be a linear function of bulk phytoplankton and cyanobacteria concentrations:

$$Chl(z) = \frac{1}{R_{Phy}} \cdot Phy(z) + \frac{1}{R_{Cya}} \cdot Cya(z). \qquad (2)$$

with $Phy(z)$ and $Cya(z)$ being vertical profiles of bulk phytoplankton and cyanobacteria concentrations in carbon units, and $R_{Phy}$ and $R_{Cya}$ the respective C:Chl ratios. Phytoplankton and cyanobacteria concentrations are not only a function of depth (z), but also vary horizontally and temporally, resulting in a temporally and spatially varying chlorophyll field Chl(x, y, z, t).

Radiative heating of water is accounted for by an internal source of heat in the temperature equation, proportional to the vertical derivative of the light field $I(z)$:

$$\frac{\partial T}{\partial t} = \frac{1}{\rho \cdot c_p} \cdot \frac{\partial I(z)}{\partial z} \qquad (3)$$

with $\rho$ as the density and $c_p$ the specific heat capacity of seawater. Thereby, it is assumed that the total absorbed light is converted into heat. Biological fluorescence mostly converts into heat, and the absorbed energy stored in biomass is generally small and can be neglected (Lewis et al., 1983, and references therein).





**Table 1.** Experiment names and descriptions.

| Experiment name | Description |
|---|---|
| PHY_ONLY | light absorption by bulk phytoplankton only (cyanobacteria are set to transparent) |
| PHY_CYA | light absorption by bulk phytoplankton and cyanobacteria |
| PHY_CYAx2 | light absorption by bulk phytoplankton and cyanobacteria, with doubled chlorophyll content of cyanobacteria |
| CTRL | globally uniform attenuation depth (17 m) |

It has to be noted that the phytoplankton-dependent light attenuation scheme (Equation 1) is not part of the default standard model MPIOM (Jungclaus et al., 2013). The standard model version, e.g., used for the fifth phase of the Coupled Model Intercomparison Project (CMIP5), applies a globally uniform exponential profile of light $I(z)$ instead (Paulson and Simpson, 1977):

$$I(z) = I_0 \cdot f_{blue} \cdot exp(-k_{blue} \cdot z) \tag{4}$$

The attenuation coefficient $k_{blue}$ of 0.06 m$^{-1}$ (attenuation depth: 17 m) and the blue water fraction $f_{blue}$ of 0.41 roughly correspond to the Jerlov optical water type 1A (Jerlov, 1976; Kara et al., 2005). The attenuation depth of 17 m is in the order of magnitude as often used in climate models (Patara et al., 2012, and references therein). Radiative heating is formulated as shown in Equation 3.

## 2.3 Experimental setup

Three experiments are conducted (Table 1), which all use the phytoplankton-dependent attenuation scheme (Equation 1). In the first experiment, PHY_ONLY, only light absorption by bulk phytoplankton is accounted for. Cyanobacteria are set to transparent with respect to shortwave radiation in the physical model ($1/R_{cya} = 0$ in Equation 2). For bulk phytoplankton, the C:Chl ratio is set to $R_{Phy} = 60$ mg C (mg Chl)$^{-1}$ as is used in HAMOCC (Ilyina et al., 2013). This value lies in the middle of the observed range, which spans values from about 12 to more than 200 mg C (mg Chl)$^{-1}$, depending on species, light conditions, nutrient limitation, and temperature (e.g., Taylor et al., 1997).

In the second experiment, PHY_CYA, we include cyanobacteria in addition to bulk phytoplankton in affecting the radiative heating in the water column. For cyanobacteria, the C:Chl ratio is set to $R_{Cya} = 120$ mg C (mg Chl)$^{-1}$. Cyanobacteria generally contain less chlorophyll than other phytoplankton groups (e.g., Berman-Frank et al., 2001; Carpenter et al., 2004). The chosen value lies in the middle of the observed range for *Trichodesmium* of $\sim 40$ to 200 mg C (mg Chl)$^{-1}$ (e.g., Berman-Frank et al., 2001; Carpenter et al., 2004).

Finally, because the light absorption strength of cyanobacteria is not well constrained, we consider a sensitivity experiment, PHY_CYAx2. In this experiment, we double the chlorophyll content per cyanobacteria biomass, i.e. setting the value of R$_{Cya}$ to 60 mg C (mg Chl)$^{-1}$ which is the value also applied for bulk phytoplankton. The value lies in the lower range of observed



values for *Trichodesmium* (that means in the upper range of observed chlorophyll contents). This experiment allows us to test the sensitivity and linearity of the results to this prescribed parameter, and to assess the upper limit of the effects induced by cyanobacteria light absorption.

It has to be noted that the different C:Chl ratios for cyanobacteria are only applied for calculating the radiative heating in the physical model. In the biogeochemical model, in all experiments the same self shading effect is applied for light limitation of photosynthesis for bulk phytoplankton and cyanobacteria ($R_{Phy} = R_{Cya} = 60$ mg C (mg Chl)$^{-1}$) in order to exclude direct effects on cyanobacteria growth.

All three experiments are started from a preindustrial simulation CTRL in steady state. CTRL uses the standard model version with globally uniform light attenuation (Equation 4). All three experiments are integrated for 300 years with prescribed preindustrial atmospheric $CO_2$ volume mixing ratio (284.7 ppm). For all analyses the mean of the last 100 years is evaluated, in which the upper ocean ($\sim$500 m) shows no considerable drifts in temperature anymore. In order to assess the added impact of including prognostic cyanobacteria in addition to bulk phytoplankton in affecting the attenuation depth of shortwave radiation, we substract the mean state PHY_ONLY from PHY_CYA (and PHY_ONLY from PHY_CYAx2, respectively). We furthermore compare the model states of the three experiments against CTRL. CTRL spans 800 years and is consulted in some analyses as an estimate of the internal variability of the model.

## 3 Global distribution of cyanobacteria and chlorophyll

Cyanobacteria occur between 40° S and 40° N (see Figure 1 for PHY_CYA) where they constitute an important fraction of total phytoplankton biomass ($\sim$18 %, locally up to 90 %). The restriction of cyanobacteria's major habitat to the tropical and subtropical ocean is in agreement with in situ and satellite-based observations (Westberry and Siegel, 2006; Breitbarth et al., 2007; Bracher et al., 2009; Luo et al., 2012). The ecological niche of cyanobacteria is mainly determined by their ability to fix $N_2$, their positive buoyancy, a high optimum temperature, and a strong iron limitation (Paulsen et al., 2017). The global depth-integrated $N_2$ fixation rate of 116.6 Tg N yr$^{-1}$ (in PHY_CYA) lies within the range of reported estimates of $\sim$80–200 Tg N yr$^{-1}$ (e.g., Karl et al., 2002; Großkopf et al., 2012). The simulated maximum annual mean surface concentrations reach values of $\sim$400 mg C m$^{-3}$ (depth-integrated $\sim$700 mg C m$^{-2}$), and surface fixation rates of $\sim$300 μmol N m$^{-3}$ d$^{-1}$ (depth-integrated $\sim$5000 μmol N m$^{-2}$ d$^{-1}$). The major fraction of biomass and $N_2$ fixation ($\sim$85 %) is thereby located in the upper 20 m. The orders of magnitudes of biomass and fixation rates as well as the large scale spatiotemporal patterns are in agreement with observations (Luo et al., 2012). In the North Atlantic subtropical gyre, the model is characterized by an underestimation of cyanobacteria concentrations and $N_2$ fixation rates in comparison to observations. In the eastern tropical Pacific, on the other hand, concentrations and fixation rates are probably somewhat overestimated by the model. For a detailed description and model evaluation, see Paulsen et al. (2017).

The simulated annual mean distribution of chlorophyll (Figure 1c), derived from bulk phytoplankton and cyanobacteria (PHY_CYA) qualitatively reproduces the main global patterns of chlorophyll *a* estimates deduced from satellite measurements of ocean color (e.g., SeaWiFS Project, 2003; Gregg et al., 2005). For the comparison, it is assumed that the general large





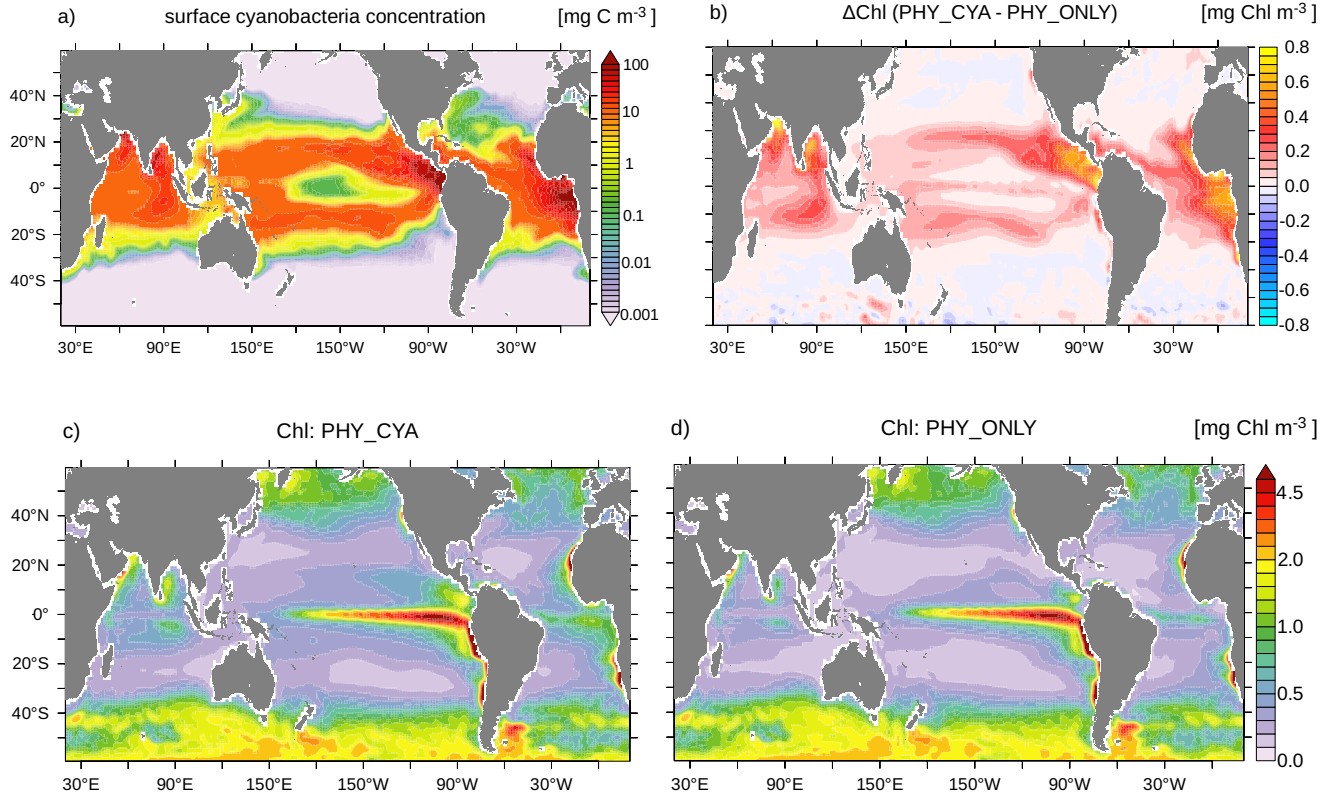

**Figure 1.** a) Climatological annual mean surface (0–22 m) cyanobacteria concentrations [mg C m$^{-3}$] in PHY_CYA. Note the nonlinear color scale. b) Difference in the climatological annual mean surface chlorophyll (0–22 m) [mg Chl m$^{-3}$] between PHY_CYA and PHY_ONLY. c) Climatological annual mean global surface (0–22 m) chlorophyll concentrations [mg Chl m$^{-3}$] as used for the light attenuation scheme in the physcial model in PHY_CYA, and d) in PHY_ONLY.

scale patterns and order of magnitudes of the concentrations have not considerably changed between the preindustrial and the satellite era. High simulated values of chlorophyll occur in high latitudes north and south of 40° latitude (annual mean ∼0.5–3 mg Chl m$^{-3}$) as well as in the nutrient rich upwelling regions at the equator and along the eastern boundaries of the ocean basins (annual mean ∼1–5 mg Chl m$^{-3}$). In the equatorial Pacific, chlorophyll values are overestimated by the

5  model, due to too high biological production caused by too strong upwelling (Ilyina et al., 2013). Low chlorophyll values, on the other hand, are present in the oligotrophic areas of the downwelling regions in the central subtropical gyres (annual mean ∼0.1–0.3 mg Chl m$^{-3}$). The simulated values in these regions are slightly higher than satellite estimates of chlorophyll (0.03–0.2 mg Chl m$^{-3}$, e.g., Gregg et al., 2005).

The contribution of cyanobacteria to annual mean surface chlorophyll becomes visible in the comparison between experi-

10  ment PHY_CYA (Figure 1c) and PHY_ONLY (Figure 1d) shown in Figure 1b, where in PHY_ONLY, cyanobacteria are set to transparent with respect to shortwave radiation. In the tropical and subtropical ocean the contribution of cyanobacteria to



surface chlorophyll is in the order of 0.1–0.3 mg Chl m$^{-3}$, regionally up to 0.8 mg Chl m$^{-3}$. This represents up to 80 % of the total surface chlorophyll concentrations in certain regions (e.g, in the eastern equatorial Atlantic and Pacific). The large contribution of cyanobacteria to surface chlorophyll in certain regions can be explained, first, by their capability to grow in nitrate depleted areas (in contrast to bulk phytoplankton), and second, by the fact that they are concentrated at the surface due to their

positive buoyancy (in contrast to neutrally buoyant bulk phytoplankton, which shows a deep chlorophyll maximum at roughly 50–100 m in large parts of the tropical and subtropical ocean in summer months). In bloom conditions, cyanobacteria biomass is often confined to the upper first meter in the real ocean. In the model, the first layer is, however, 12 m thick. If the simulated chlorophyll content of cyanobacteria of the first model layer with an annual mean concentration of up to 0.9 mg Chl m$^{-3}$ was confined more to the surface (e.g. to the upper first meter) this would refer to a value of 10.8 mg Chl m$^{-3}$, which lies within

the range of reported values for *Trichodesmium* blooms ($\sim$0.8–40 mg Chl m$^{-3}$; e.g., Subramaniam et al., 2001; Westberry and Siegel, 2006; Mohanty et al., 2010). Overall, the simulated values of cyanobacteria chlorophyll concentrations are thus in a plausible order of magnitude.

In experiment PHY_CYAx2, the patterns of cyanobacteria and chlorophyll are consistent with PHY_CYA (not shown). The chlorophyll values are, however, roughly doubled. Mean cyanobacteria chlorophyll values reach now 1.6 mg Chl m$^{-3}$ instead

of 0.8 mg Chl m$^{-3}$ as in PHY_CYA. Blooms would have a chlorophyll concentration of 21.6 mg Chl m$^{-3}$ if confined to the first meter, which also lies within the reported range.

## 4    Effects of cyanobacteria light absorption on the mean tropical climate state

### 4.1    Effects on ocean temperature and mixed layer depth

Including light absorption by cyanobacteria in addition to bulk phytoplankton (experiment PHY_CYA compared to PHY_ONLY)

has a warming effect on SST in some limited areas of the tropical ocean (Figure 2a). In largest parts, in contrast, a cooling effect on SST dominates. The mean cooling effect in the tropical area (20° S–20° N) is small (0.07 K). Regionally, however, the negative SST anomalies reach values of up to 0.5 K in PHY_CYA which is significantly larger than the internal variability of the model (dotted areas in Figure 2b show anomalies larger than 90 % significance according to the Student's *t*-test). The strongest negative anomalies occur at the equator and in the eastern boundary upwelling systems off the South American and

African continents. The surface cooling can be explained by the following mechanism. The presence of cyanobacteria increases light absorption in the upper layers. Radiative heating is thus more confined to the surface, causing a significant shoaling of the mixed layer depth (MLD) by up to 10 m (Figure 2d) and a shoaling of the thermocline depth (shown for the equatorial Pacific in Figure 3). At the same time, deeper layers receive less radiation, which leads to a decrease in subsurface temperature (Figure 2c), in the global zonal mean by up to 0.5 K (Figure 4). The cold signal reaches below the MLD and down to the

thermocline (defined here as the depth of the 20°C isotherm). This subsurface water feeds the shallow overturning cells and is transported equatorward along the thermocline (which is equivalent with the pycnocline in the Tropics). At the equator, the relatively cooler subsurface water is upwelled, where it outweighs the direct heating effect due to local cyanobacteria light absorption at the surface. The upwelled water is spread laterally via the poleward surface Ekman transport and leads to a cool-




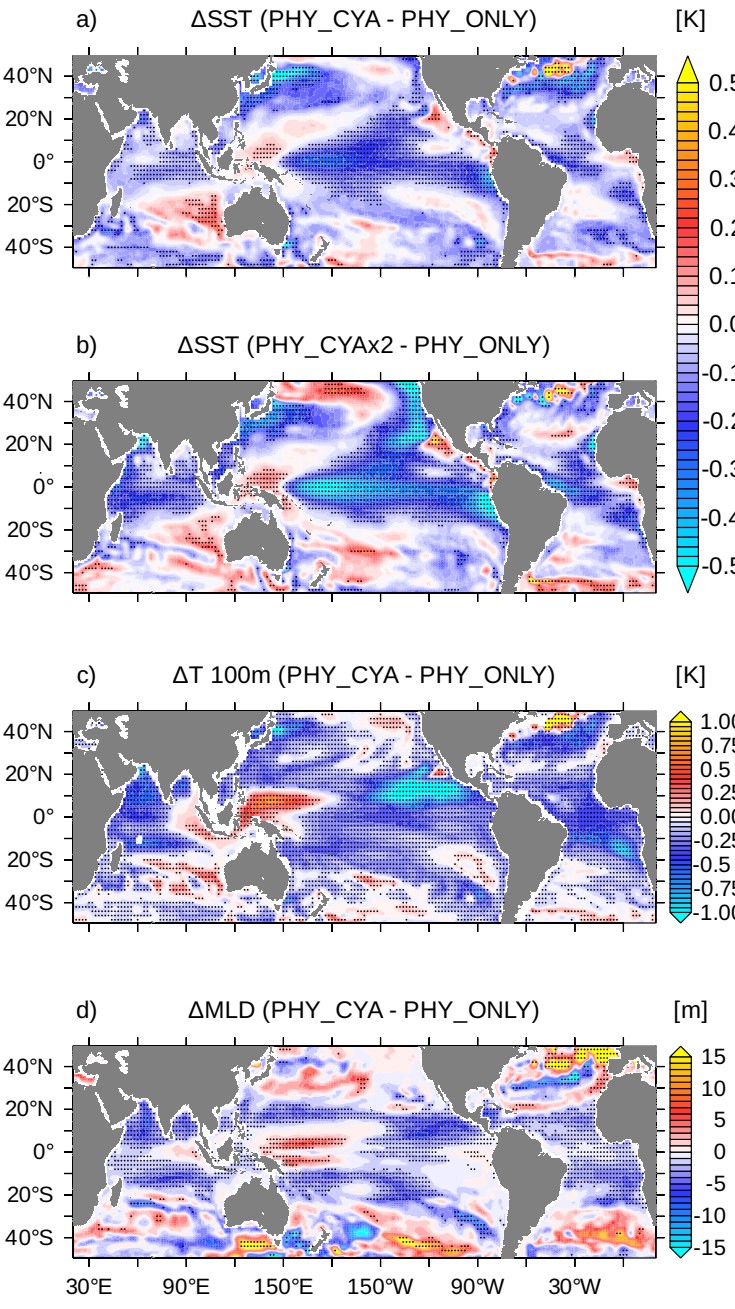

**Figure 2.** a) Difference in the climatological annual mean SST [K] between PHY_CYA and PHY_ONLY and b) between PHY_CYAx2 and PHY_ONLY. c) Difference in the climatological annual mean temperature at a depth of 100 m [K], and d) mixed layer depth [m] between PHY_CYA and PHY_ONLY. Dotted areas show anomalies larger than 90 % significance (Student's *t*-test).





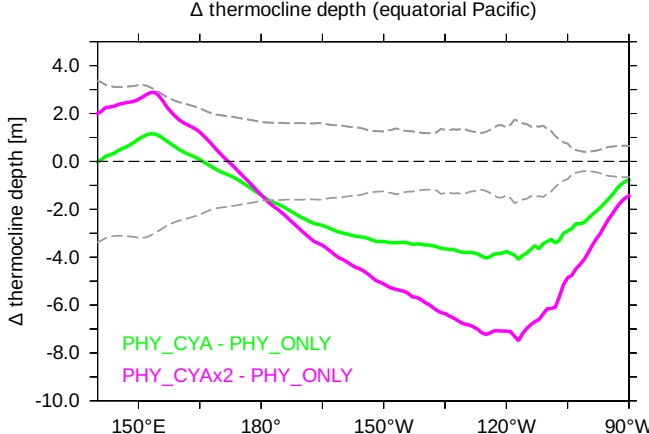

**Figure 3.** Difference in the climatological mean thermocline depth [m] between PHY_CYA and PHY_ONLY (green) and PHY_CYAx2 and PHY_ONLY (purple). To provide a sense for the internal variability of the model, the grey dashed lines show ± two times the standard deviation of eight 100 year periods of CTRL.

ing in large parts of the tropical and subtropical ocean. Hence, instead of an expected surface heating induced by the presence of cyanobacteria, the advective process bringing relatively cooler subsurface water to the surface dominates. In addition, the atmospheric cooling above respective regions spreads throughout the troposphere and enhances the surface cooling in other regions due to an enhanced heat flux directed from the ocean into the atmosphere (not shown).

The cold upwelled subsurface water originates from regions where cyanobacteria are abundant and shade the deeper layers (Figure 1a,b). These regions mainly comprise two different regimes. The first regime includes the downwelling regions of the subtropical gyres in which the cooler water is subducted within the shallow meridional overturning cells and then transported equatorward along the thermocline. The second regime includes the eastern boundary upwelling regions at the margins of the subtropical gyres (western Africa north and south of the equator, and eastern tropical Pacific, north of the equator). Here,

overall high cyanobacteria concentrations are present, which strongly shade and hence cool the subsurface layers. Also from here, the water partly ends up in the shallow meridional overturning cells and the equatorial upwelling system, contributing to the cold SST signal seen at the equator. Due to the complexity of the circulation system and the exchange pathways between the tropical and subtropical ocean, it is difficult to separate the contributions from the subtropical gyres and from its margins to the cooling signal.

Besides the SST decrease, which dominates in largest parts of the tropical and subtropical ocean in PHY_CYA compared to PHY_ONLY, there are some limited regions with a significant annual mean SST increase as mentioned above (Figure 2a). These regions include the eastern equatorial Atlantic and the eastern equatorial Pacific adjacent to the coast. In these regions, the direct heating by high cyanobacteria concentrations together with the decrease in upwelling strength (as will be discussed in Section 4.3), outweighs the upwelling effect of cooler subsurface water and results in a surface warming effect. Whether





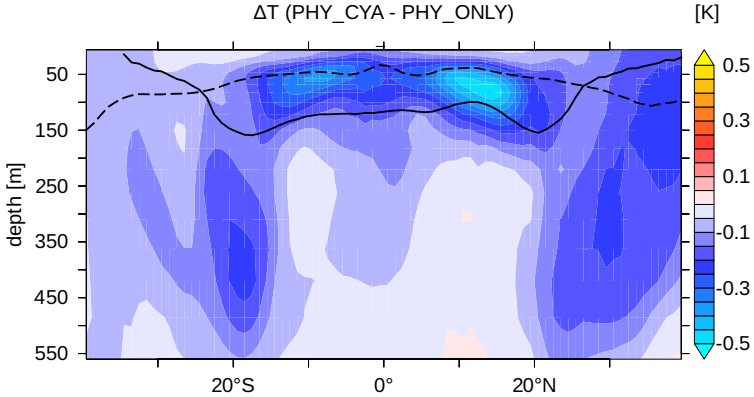

**Figure 4.** Difference in the climatological global zonal mean temperature [K] in the upper 550 m between PHY_CYA and PHY_ONLY. The dashed black line shows the global zonal mean MLD and the solid black line the global zonal mean thermocline depth (defined as the depth of the 20°C isotherm) in PHY_ONLY.

light absorption by cyanobacteria leads locally to a mean surface warming or cooling thus depends on two opposing effects: the local heating due to light absorption and the non-local cooling due to upwelling of cooler subsurface water. Which process dominates, strongly depends on the region and on the local cyanobacteria concentrations. Two other regions with significant positive SST anomalies are the western Pacific warm pool and the eastern subtropical Indian Ocean. Here, not the local effect

of high light absorption is causing the anomalies, but probably changes in the circulation system (Section 4.3).

In the sensitivity experiment, PHY_CYAx2, the patterns of change in SST relative to PHY_ONLY are consistent with PHY_CYA relative to PHY_ONLY (compare Figures 2a and b, respectively). The magnitudes of the anomalies, both the positive and negative ones, are, however, strongly enlarged by roughly a factor of two (e.g., equatorial Pacific: ∼0.6 K in PHY_CYAx2 compared to ∼0.3 K in PHY_CYA). The higher chlorophyll content of cyanobacteria leads to a stronger local

warming in regions where they are present. But at the same time this also leads to a stronger shading of the subsurface water, which enhances the subsurface cooling and which brings cooler water to the surface. The resulting patterns of the net effect (cooling or warming) are consistent between the two experiments. The consistency of the anomaly patterns between PHY_CYAx2 and PHY_CYA (relative to PHY_ONLY) also counts for most other quantities which are analyzed in this study. Thus, in the following, only the anomaly maps for experiment PHY_CYA are shown. The respective numbers for PHY_CYAx2

are stated in the text.

## 4.2 Effects on wind patterns and precipitation

The cooling of the cold tongue in the east Pacific and heating of the warm pool in the west Pacific in experiment PHY_CYA compared to PHY_ONLY intensifies the zonal SST gradient, implying a strengthening of the Walker circulation (Figure 5a). The ascent of the air over the warm pool is enhanced west of 150° E by roughly 6 % (13 % in PHY_CYAx2) (vertical





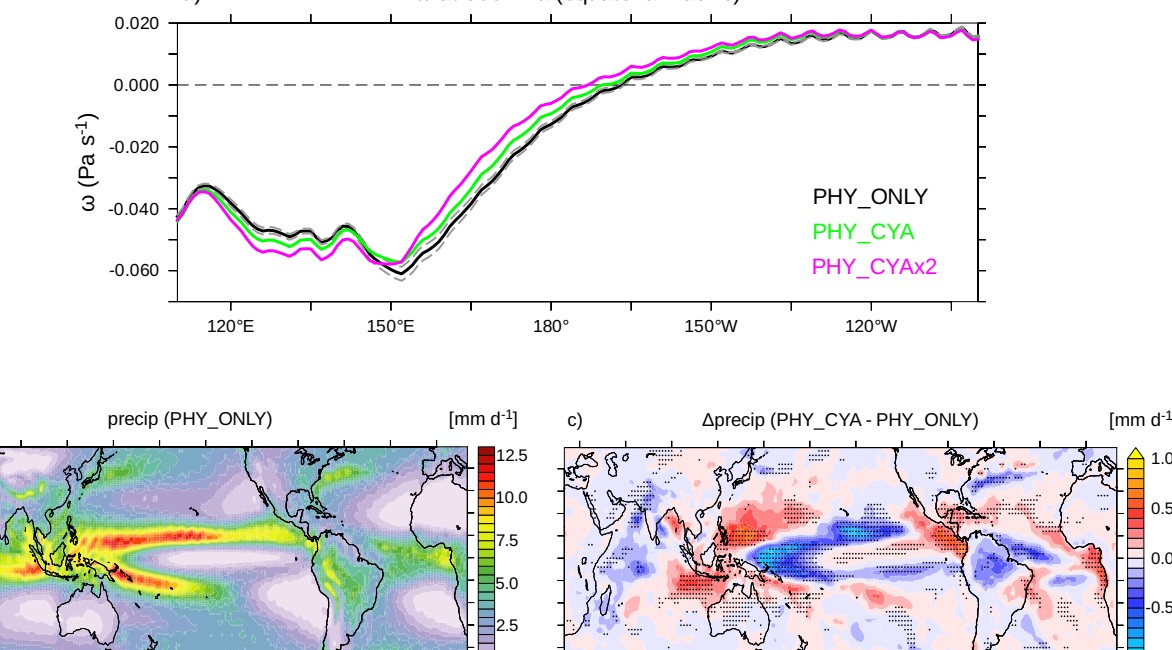

**Figure 5.** a) Climatological mean vertical velocity $\omega$ [Pa s$^{-1}$] at 500 hPa above the equatorial Pacific (10° S–10° N) in PHY_ONLY (black, $\pm$ two times the standard deviation of eight 100 year periods of CTRL in dashed grey), PHY_CYA (green) and PHY_CYAx2 (purple). b) Climatological annual mean precipitation [mm d$^{-1}$] in PHY_CYA. c) Difference in the climatological annual mean precipitation [mm d$^{-1}$] between PHY_CYA and PHY_ONLY. The dotted areas show anomalies larger than 90 % significance (Student's $t$-test).

velocity $\omega$ at 500 hPa averaged from 120–140° E), and the descent of air in the east by roughly 10 % (17 % in PHY_CYAx2) (averaged from 120-160° W, respectively). The equatorial westward winds are intensified west of 160° W, enhancing the westward windstress on the ocean (not shown). Furthermore, the transition zone between convection and subsidence of the Walker circulation is shifted eastward by ∼3° longitude (∼6° longitude in PHY_CYAx2) (Figure 5a). This is a significant

5    change, larger than the internal variability of the model, which is estimated as $\pm$ two times the standard deviation of eight 100 year periods of experiment CTRL (dashed grey lines in Figure 5a). In other words, the convection gets stronger, but more confined to the western part of the Pacific basin.

The strong negative SST anomaly at the equator generally reduces the meridional SST gradient between the equatorial zone and the Extratropics (Figure 2a). This results in weaker trade winds and a reduction of the Hadley circulation (not shown).



At the same time, the Hadley cells are slightly expanded polewards which leads to a northward shift of the western boundary currents (Kuroshio Current and Gulf Stream) as visible in the dipole SST pattern (Figure 2a).

The strongest effect on precipitation can be seen over the western equatorial Pacific (Figure 5c). Here, the strong decrease in equatorial SST at $160°$ E (Figure 2a) and the zonal displacement of the Walker circulation (Figure 5a) lead to a change in the location and strength of convection and precipitation. Precipitation significantly decreases with a maximum centered at $\sim 160°$ E by up to 1.0 mm d$^{-1}$ (up to 1.6 mm d$^{-1}$ in PHY_CYAx2), and extending off-equator in latitudinal bands at about $10°$ S and $10°$ N towards the east. A significant decrease is also seen above the cold SST anomalies in the western Indian Ocean, as well as the western tropical Atlantic and over the Amazon region. In the eastern parts of the ocean basins, the eastern equatorial Atlantic, Pacific and Indian Ocean, relatively warmer surface water, on the other hand, increases convection and hence precipitation by up to $\sim 0.8$ mm d$^{-1}$ ($\sim 1.0$ mm d$^{-1}$ in PHY_CYAx2).

### 4.3 Effects on ocean circulation

The weakening of the Hadley cells in PHY_CYA (and PHY_CYAx2) compared to PHY_ONLY implies a weakening of the wind driven ocean circulation. The barotropic streamfunction $\Psi$, which describes the large-scale horizontal ocean circulation, significantly decreases in magnitudes both in the subtropical- as well as in the equatorial gyres (Figure 6a,b). The dipole anomaly patterns within the subtropical gyres indicate a poleward shift of its boundaries. This feature is most dominant in the North Atlantic, where the weakening in the southern part and strengthening in the northern part reflect a northward shift of the position of the Gulf Stream and the North Atlantic Current, respectively. The northward shift is also visible in the strong negative temperature anomaly which is present at the surface (Figure 2a) and reaches down to greater depth (Figure 4).

In the North Pacific, the increase of annual mean southward windstress along the Pacific coast (not shown) reduces the warm water transport of the Kuroshio Current. Together with the slight shift of the western boundary current northward, this leads to the strong negative SST anomaly, which reaches into the interiour of the ocean basin (Figure 2a).

In the Indian Ocean, both the equatorial and the subtropical gyres are generally weaker in PHY_CYA compared to PHY_ONLY and the boundary between them is slightly shifted southwards (Figure 6a,b). The significant reduction in strength of the Indian Ocean subtropical gyre is accompanied by a weakening of the so-called Southern Hemisphere "supergyre" (e.g., Speich et al., 2007), which extends into the South Atlantic. The associated reduced transport of warm water from the Indian Ocean into the South Atlantic ocean is visible in the cold SST signal around the tip of South Africa (Figure 2a). This cooling effect enhances the negative SST signal which occurs at the eastern margin of the South Atlantic due to the upwelling of colder subsurface water in the Benguela Upwelling system.

In the Atlantic Ocean, the reduction of the trade winds and the related windstress on the ocean (not shown) reduces equatorial divergence. The related weakening of the overturning cells slightly dampens equatorial upwelling and reduces downwelling in the subtropical gyres (Figure 6c). Thus, the atmospheric response to the SST anomaly pattern does not amplify the negative equatorial SST signal, but rather dampens it, by transporting less cold subsurface water to the surface. In combination with the local heating due to high cyanobacteria concentrations, this results in a positive SST signal at the eastern boundary (Figure 2a). In the Pacific Ocean, a similar behavior of weaker trade winds and equatorial upwelling (Figure 6d) can be seen in the eastern



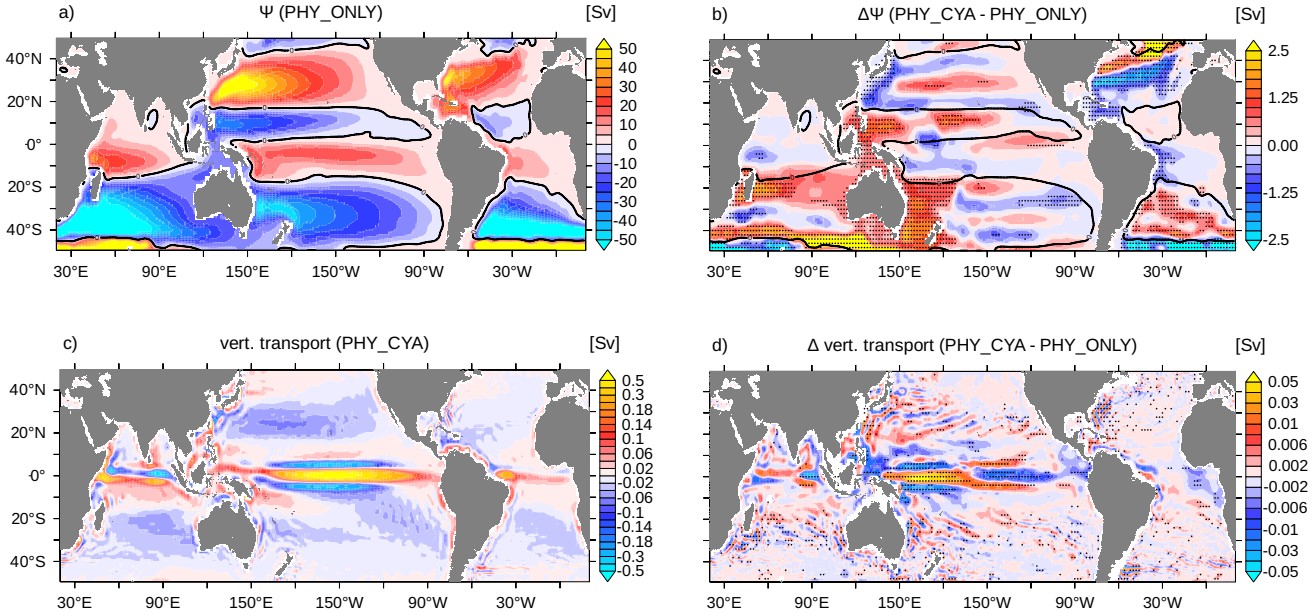

**Figure 6.** a) Climatological mean barotropic streamfunction $\Psi$ [Sv] in PHY_ONLY. The zero-isoline is overlaid in black. b) Difference in the climatological mean $\Psi$ [Sv] between PHY_CYA and PHY_ONLY. The zero-isoline of PHY_ONLY is overlaid in black. c) Climatological mean vertical transport in the upper 100 m [Sv] in PHY_ONLY. Note the non-linear color scaling with 0.01 Sv between -0.2 and 0.2 Sv and 0.1 Sv for the remaining range. d) Difference in the climatological annual mean vertical transport in the upper 100 m [Sv] between PHY_CYA and PHY_ONLY. Note the non-linear color scaling with 0.001 Sv between -0.01 and 0.01 Sv and 0.01 Sv for the remaining range. The dotted areas in b) and d) show anomalies larger than 90 % significance (Student's *t*-test).

part of the basin (east of $160°$ W). In the western part, on the contrary, the strengthening of the westward surface current related to the change in the Walker circulation increases equatorial upwelling. Here, the atmospheric feedback intensifies the surface cooling effect. This explains the larger SST signal in the western central Pacific compared to the east (Figure 2a). In summary, the atmospheric feedback amplifies the SST signal in the western Pacific by enhancing the upwelling strength, and dampens it,
5  on the other hand, in the eastern Pacific and the Atlantic Ocean by reducing the upwelling strength.

## 4.4 Feedback on phytoplankton abundance

By comparing the simulations with and without cyanobacteria light absorption we quantify how cyanobacteria induced changes in the physical ocean state feed back on the growth conditions of cyanobacteria and bulk phytoplankton itself. Globally integrated, the biogeochemical mean states in PHY_CYA, PHY_CYAx2 and PHY_ONLY do not differ notably from each other
10  in quantities like the total cyanobacteria and phytoplankton biomass, nitrogen fixation and primary production. Regionally, however, significant changes in the concentrations of bulk phytoplankton and cyanobacteria occur (Figure 7). Thereby, phy-





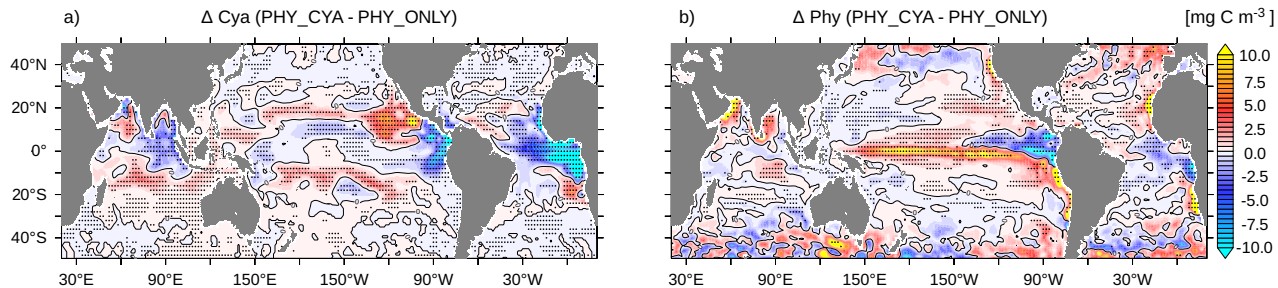

**Figure 7.** Change in a) climatological annual mean surface (0-22 m) cyanobacteria concentrations [mg C m$^{-3}$] and b) surface bulk phytoplankton concentrations [mg C m$^{-3}$] between PHY_CYA and PHY_ONLY. The zero-isolines are overlaid in black. The dotted areas show anomalies larger than 90 % significance (Student's *t*-test).

toplankton concentrations, both of cyanobacteria and bulk phytoplankton, do not uniformly decrease or increase, but show a more complex pattern of change. Hence, we cannot generally refer to a positive or negative feedback between ocean physics and cyanobacteria light absorption. A positive feedback means that induced changes in ocean circulation and temperature result in an increase in cyanobacteria biomass, which further enhances the initial perturbation - i.e., the increase in light absorption

strength. This is indeed the case in the subtropical bands of the Pacific ocean. Here, the cyanobacteria abundance results in cooling of equatorial SST (Figure 2a), which shifts the Walker circulation westward (Figure 5a), implying an increase in upwelling strength in the western equatorial Pacific (Figure 6d). The related increased abundance of nutrients at the surface is spread poleward via Ekman transport and promotes growth of cyanobacteria (and partially also bulk phytoplankton) in the subtropical bands at about ∼10–20° S and ∼10–20° N (Figure 7), closing the positive feedback loop. In the eastern tropical

Atlantic and Indian Ocean, as well as the eastern equatorial Pacific, the decrease in upwelling strength and the increase in SST above the optimum temperature of 28°C impair cyanobacteria growth conditions and results in a strong decline in cyanobacteria concentrations of up to 25 % (Figure 7a). In experiment PHY_CYAx2, the decline in these regions reaches values of up to 60 % (not shown). The cyanobacteria decline dampens the effect of light absorption, thus constituting a negative feedback. The strong increase in bulk phytoplankton in the equatorial Pacific, as well as in the eastern boundary upwelling areas (Figure 7b),

also rather dampens the cooling effect due to the large increase in surface light absorption directly at the upwelling sites. In summary, the net effect of including cyanobacteria's effect on light absorption on cyanobacteria and phytoplankton growth is regionally different and results from an interplay of different physical and biogeochemical processes.

## 5  Effects of cyanobacteria light absorption on seasonal dynamics and interannual variability of SST

Including cyanobacteria light absorption modifies not only the mean climate state but also the seasonal SST amplitude as

well as interannual SST variability. We find an increase in the amplitude of the climatological mean seasonal cycle of SST in PHY_CYA compared to PHY_ONLY in large parts of the tropical and subtropical ocean, regionally by up to ∼25 % (Figure 8)

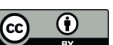



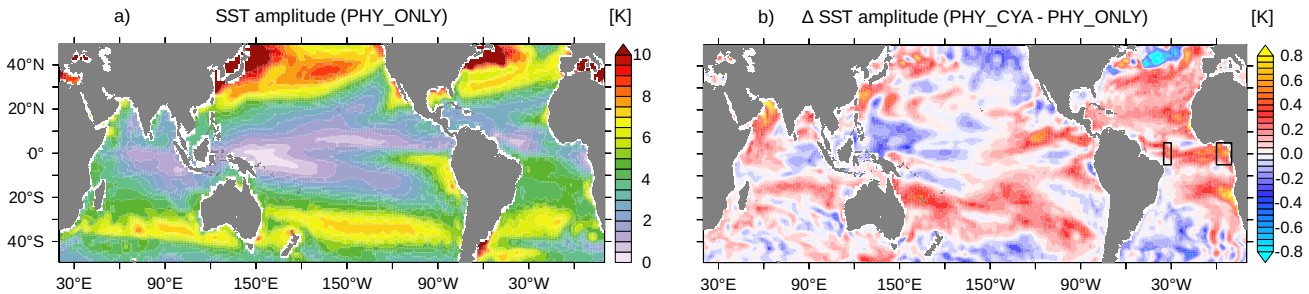

**Figure 8.** a) Climatological mean amplitude of the seasonal cycle of SST [K] in PHY_ONLY. b) Difference in the climatological mean amplitude of the seasonal cycle of SST [K] between PHY_CYA and PHY_ONLY. The black boxes display the locations discussed in Section 5 (Figure 9).

(in PHY_CYAx2 up to ∼50 %). In large parts, this increase improves the model performance in comparison to the observed amplitude of the seasonal cycle (Reynolds et al., 2007, not shown). The increase in the simulated amplitude is characterized by either an increase in the maximum annual monthly mean temperature (caused by the direct heating of local cyanobacteria biomass) or by a decrease of the minimum annual monthly mean temperature (caused by the indirect cooling due to the

upwelling of subsurface water and its subsequent lateral transport), or by a combination of both. Both processes have their own – partly interdependent – seasonal dynamics. The changes in temperature and upwelling strength furthermore feed back on the cyanobacteria abundance itself. We exemplarily select two locations (the western and eastern tropical Atlantic) and show the climatological seasonal cycle of SST and surface cyanobacteria biomass (Figure 8b, black boxes) to illustrate that, depending on the region, different factors play a role.

In the western Atlantic equatorial upwelling region, a stronger surface cooling prevails year-round in PHY_CYA (and PHY_CYAx2) compared to PHY_ONLY (Figure 9a) due to upwelling of relatively cooler subsurface water. The decrease is stronger from July to September (due to low cyanobacteria concentrations and hence surface light absorption), resulting in an increased amplitude of the seasonal cycle of 13.5 % in PHY_CYA (19.7 % in PHY_CYAx2) which slightly impairs the model results in comparison to the observed amplitude of the seasonal cycle at this location (Reynolds et al., 2007). Cyanobac-

teria concentrations are lower in PHY_CYA (and PHY_CYAx2) compared to PHY_ONLY at the beginning of the year due to weaker upwelling (Figure 9b).

      In the eastern equatorial Atlantic (Gulf of Guinea), maximum temperatures between December and March are increased in PHY_CYA (and PHY_CYAx2) compared to PHY_ONLY due to the surface heating effect of the high local cyanobacteria concentrations which prevail in this period. From May to October, on the other hand, the cyanobacteria concentrations – and

hence the surface warming effect – decrease and, instead, the upwelling of colder subsurface water dominates and results in a slight decrease in SST. In total, the amplitude of the seasonal cycle of SST is enhanced (17.0 % in PHY_CYA, 21.1 % in PHY_CYAx2, Figure 9c). This brings it slightly closer to the observed amplitude (Reynolds et al., 2007, not shown) which



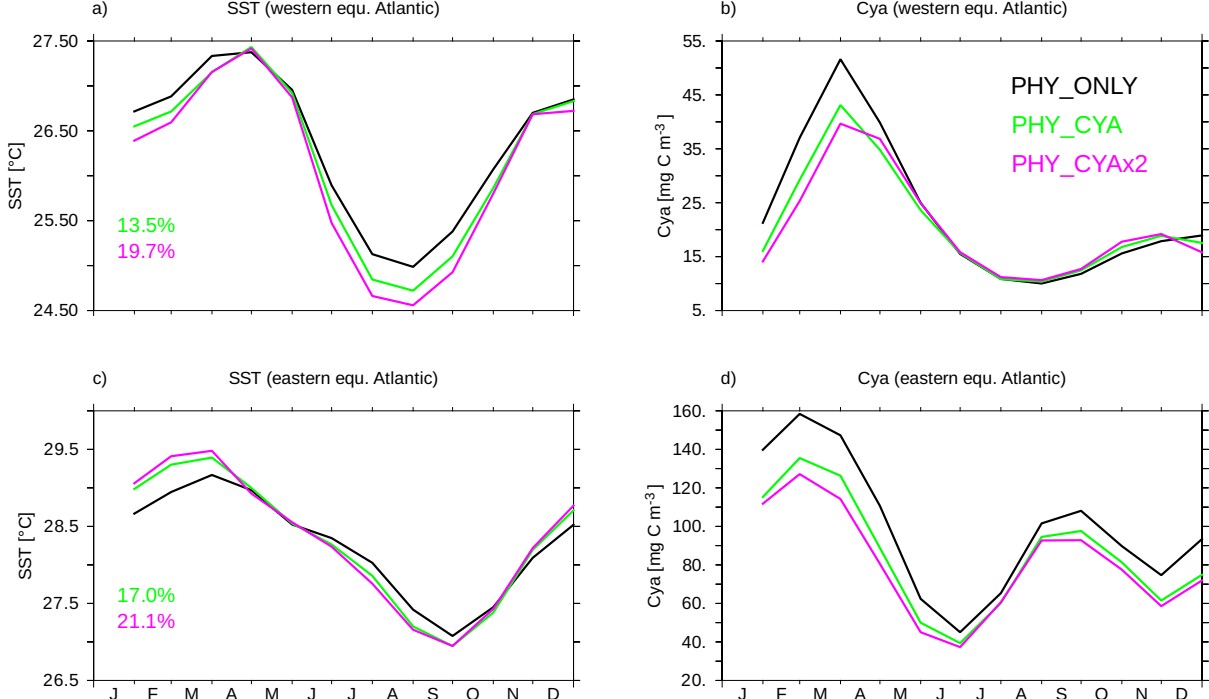

**Figure 9.** a) and c): Climatological monthly mean seasonal cycle of SST [°C] in the western (top) and eastern (bottom) tropical Atlantic in PHY_ONLY (black), PHY_CYA (green), and PHY_CYAx2 (purple). The displayed numbers show the change in the amplitude in percent. b) and d): Climatological mean seasonal cycle of surface cyanobacteria concentrations [mg C m$^{-3}$] in the western (top) and eastern (bottom) tropical Atlantic in PHY_ONLY (black), PHY_CYA (green), and PHY_CYAx2 (purple). The locations are shown as black boxes in Figure 8b.

is generally underestimated in the model in this region of the ocean. Cyanobacteria growth and concentrations are reduced in PHY_CYA (and PHY_CYAx2) compared to PHY_ONLY all year round (Figure 9d) since the upwelling strength is weaker (Figure 6d) and SST exceeds the optimum value of 28°C in the beginning of the year.

Besides the seasonal dynamics of SST, also the interannual variability of SST is affected by cyanobacteria light absorption.

5   The interannual standard deviation of SST in the equatorial Pacific (160° E-80° W,10° S-10° N) is increased by 21.8 % in PHY_CYA compared to PHY_ONLY (Figure 10a,b) which exceeds an estimate of centennial anomalies of the model (Figure 10c, dots show significant anomalies larger than ± two times the standard deviation of eight 100 year periods of CTRL). Regionally, the increase reaches values of 60 %. The Niño 3.4 index (SST anomalies averaged over the area 5° S–5° N, 170° W–120° W) is increased by 16 % (from 0.69 K in PHY_ONLY to 0.80 K in PHY_CYA). The increase in variability

10  is probably related to the mean state changes induced by the presence of cyanobacteria: first, the increased assymetry in zonal SST due to the cooling of the cold tongue (Figure 2a), and second, the shoaling of the equatorial thermocline (Figure 3). Both factors are generally proposed to increase variability of tropical SST and El Niño-Southern Oscillation (ENSO) (e.g., Collins

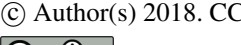



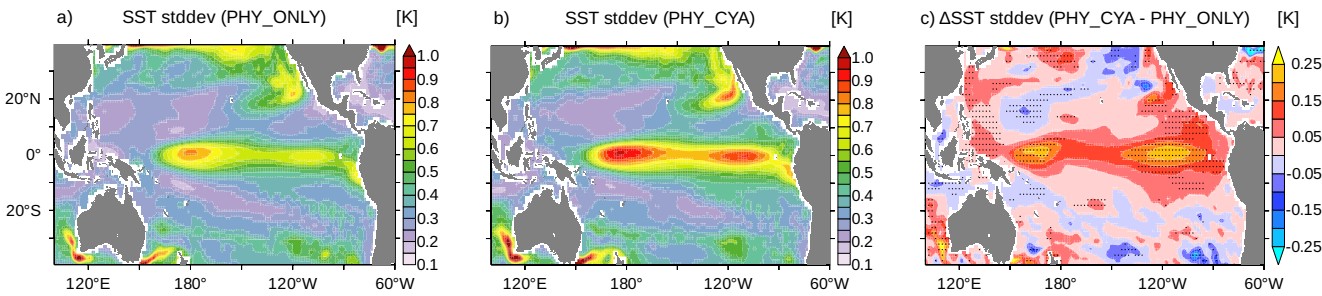

**Figure 10.** Standard deviation of SST [K] in the tropical Pacific in a) PHY_ONLY and b) PHY_CYA, and c) the difference between PHY_CYA and PHY_ONLY. The dotted areas show anomalies larger than ± two times the standard deviation of eight 100 year periods of CTRL.

et al., 2010; Meehl et al., 2001). Hence, cyanobacteria, although mainly growing off-equator, influence ENSO dynamics by remotely modifying SST and the thermocline depth at the equator.

Also in experiment PHY_CYAx2, interannual variability is increased compared to PHY_ONLY (not shown). A stronger cooling of the cold tongue (Figure 2b) and shoaling of the equatorial thermocline (Figure 3) in PHY_CYAx2 compared to
PHY_CYA would let expect an even stronger increase in variability (e.g., Collins et al., 2010; Meehl et al., 2001). The increase is, however, slightly lower (17.1 % in the area 160° E-80° W,10° S-10° N, and regionally up to 50 %). This indicates that there are other factors besides the mean state changes in zonal SST and thermocline depth which also influence the tropical Pacific interannual variability.

# 6   Discussion and synthesis

## 6.1   Discussion in the context of observations and previous model studies

Including light absorption by cyanobacteria has considerable effects on the model's tropical climate mean state and variability. The net surface cooling effect induced by cyanobacteria in large parts of the tropics which we see in our simulations is rather counterintuitive. Theoretical considerations and a one-dimensional model study (Sonntag and Hense, 2011) would let expect a local surface warming to take place due to cyanobacteria light absorption. Also satellite and in situ observations show a local
warming effect due to the presence of a cyanobacteria blooms (Kahru et al., 1993; Capone et al., 1998; Wurl et al., 2018). These observations, however, only represent snapshots of episodical heating events. Within the framework of the Earth system model used in this study, we show that on climatological time scale this local surface warming effect prevails only in limited areas. In largest areas, on the contrary, the surface heating effect of cyanobacteria acting locally, transfers into a cooling effect on the climatological large scale due to the upwelling of cooler subsurface water. Cyanobacteria, by remotely affecting regions



relevant for biogeochemistry and climate variability, thus play a role on a larger scale than indicated by local observations and one-dimensional models.

A number of previous model studies concur with the relevance of the advective redistribution of heat and also diagnosed a cooling of equatorial SST caused by the presence of phytoplankton (Nakamoto et al., 2001; Sweeney et al., 2005; Anderson et al., 2007; Gnanadesikan and Anderson, 2009; Sonntag, 2013). In an idealized regional ocean model of the North Atlantic Sonntag (2013) found that cyanobacteria locally heat the surface ocean, and non-locally cool the equatorial SST due to advective processes – a result which qualitatively agrees with our study. The global model studies of Anderson et al. (2007) and Gnanadesikan and Anderson (2009) focused on similar regions as we do in this study. The authors showed that especially the water turbidity in off-equatorial regions, more precisely in the subtropical gyres and the eastern tropical regions overlying the oxygen minimum zones, leads, in agreement with our study, to a surface cooling with similar patterns and effects on the mean climate state. The concordance about the sensitivity of the Walker and Hadley circulation as well as precipitation patterns to the changes in SST between this study and the coupled model study by Gnanadesikan and Anderson (2009) gives confidence about the robustness of our model results. Whereas Gnanadesikan and Anderson (2009) used a satellite climatology of chlorophyll, we applied a full biogeochemical model with interactive biogeophysical feedback. This ensures – in contrast to a satellite climatology of chlorophyll – 1) a temporally varying chlorophyll field that is consistent with the model circulation field (due to nutrient and temperature limitation of phytoplankton growth which is to a large degree affected by the ocean circulation), 2) vertical variations in chlorophyll concentrations, 3), most importantly, the feedback from changes in physics on chlorophyll concentrations themselves. Furthermore, we were able to identify cyanobacteria to be the phytoplankton group that is largely responsible for the light absorption in the areas which are already considered important by Gnanadesikan and Anderson (2009). Cyanobacteria, provided with the ability to fix $N_2$, inhabit exactly the regions which have a regulative effect on tropical SST due to shading of the subsurface water that enters the shallow meridional cells. Moreover, cyanobacteria are positively buoyant and hence mainly concentrated in the upper 20 m, where highest incoming radiation is present, and where effects on light absorption thus have strongest effects on temperature. Based on idealized model setups, Sonntag and Hense (2011) and Hense et al. (2017) already pointed out the potential necessity of "surface mat producers", such as positively buoyant cyanobacteria, to be included in Earth system models – a suggestion which we confirm in our study and furthermore underpin with numbers.

The cooling effect due to off-equatorial turbidity was also stressed in global ocean-only studies. By considering the zonal momentum budget integrated over the mixed layer, the authors argue that stronger light absorption and related shallower MLDs in the regions off-equator enhance the meridional transport and equatorial upwelling, which causes the simulated SST decrease at the equator (Sweeney et al., 2005; Löptien et al., 2009; Park et al., 2014). In our coupled model setup, on the contrary, we show that the atmospheric feedback on ocean circulation largely outweighs the effect on upwelling induced by the change in the MLD and results, instead, in a decrease in upwelling strength in most parts. In the coupled system, the shading and hence cooling of the subsurface water that is eventually upwelled is responsible for the cooling effect, rather than an increase in upwelling strength itself. This discrepancy between the results of the ocean-only and the coupled model setup emphasizes the importance of using an Earth system model to study biogeophysical feedbacks in order to account for interactions between the different components of the Earth system.



The fact that including the retroaction of phytoplankton on the surface ocean heat budget affects ENSO dynamics, which we find in our simulations, has already been suggested by previous model studies (e.g., Timmermann and Jin, 2002; Marzeion et al., 2005; Jochum et al., 2010). Many of the studies, however, attribute this effect to the presence of chlorophyll at the equator. Our study indicates, in agreement with Anderson et al. (2009), that especially light absorption off-equator plays a relevant role

by modifying equatorial SST and thermocline depth. As cyanobacteria are strongly contributing to the light absorption strength off-equator, they impose further complexity on ENSO dynamics which has not been taken into account in previous studies.

The use of an Earth system model with interactive biogeochemistry (instead of a chlorophyll climatology) furthermore allows us to study the feedback from phytoplankton induced changes in climate on phytoplankton growth itself. As mentioned in Section 2.3, we changed the respective C:Chl ratio only in the light attenuation scheme of the physical model. In the

biogeochemistry model we kept the same ratio to exclude direct effects on phytoplankton growth. If the different C:Chl ratios were also applied in the light limitation for photosynthesis of cyanobacteria, this would slightly change the growth rates and hence overall magnitudes of chlorophyll. The patterns, as well as the resulting effects on climate would, however, be qualitatively the same. The positive feedback (cyanobacteria promote their own growth due to the process of light absorption) which we find in some regions, such as the subtropical bands of the Pacific Ocean, is in line with idealized model studies

(Hense, 2007; Sonntag and Hense, 2011; Sonntag, 2013). In our simulations, however, the positive feedback is not acting locally via changes in temperature, as is the case in the studies mentioned above, but mainly via non-local processes, i.e. changes in the upwelling strength and hence the supply of nutrients. In large regions, these circulation changes lead to a decline in cyanobacteria concentrations in our simulation, which rather constitutes a negative feedback. Nutrient limitation of cyanobacteria growth is not included in abovementioned studies, which could explain the diversing results.

As seen in our sensitivity experiment, the patterns of change in climate properties induced by including cyanobacteria in the absorption feedback are largely independent of the prescribed absorption strength (i.e. chlorophyll content) of cyanobacteria. The magnitudes of the effects behave roughly linear: a doubling of the absorption strength roughly doubles the anomalies for many analyzed quantities in large areas. This is not obvious, since many non-linear processes are involved and the resulting local heating effect and non-local cooling effect do not necessarily need to add up to the same net effect (surface cooling or

warming). The magnitudes of the effects are very sensitive to the prescribed strength of light absorption, i.e. the chlorophyll content. Since this parameter is not well constrained, there is a considerable potential range of impact induced by cyanobacteria light absorption. Chlorophyll concentrations are derived linearly from the cyanobacteria concentrations and thus have to be seen as a rough first order approximation. In reality, chlorophyll content depends on a lot of factors, e.g., light conditions and temperature. For both applied parameters, 120 and 60 mg C (mg Chl)$^{-1}$, the changes in the climate mean state and

variability induced by including cyanobacteria are significant compared to the internal variability of the model. The value of 60 mg C (mg Chl)$^{-1}$ as applied in PHY_CYAx2, is situated in the lower range of observations (i.e. in the upper range of chlorophyll contents) (e.g., Berman-Frank et al., 2001; Carpenter et al., 2004; Sathyendranath et al., 2009). The effects in this experiment can thus be roughly seen as upper limit. On the other hand, cyanobacteria contain also pigments other than chlorophyll, like phycocyanin, which also absorb shortwave radiation in the visible range (e.g., Navarro Rodriguez, 1998), but

are not accounted for in the model parameterization.





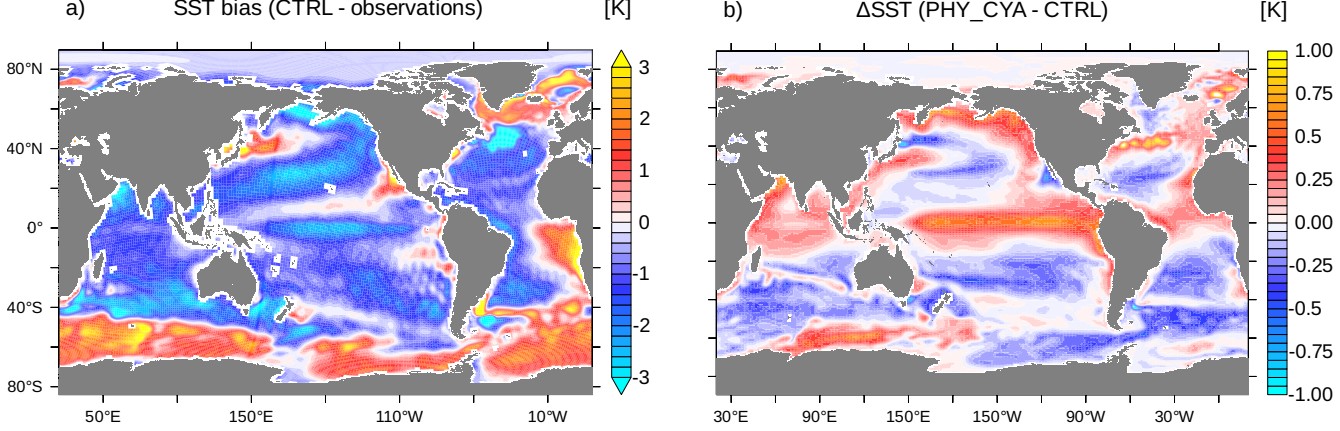

**Figure 11.** a) SST [K] bias of CTRL compared to observations (PHC3 climatology). b) Difference in SST [K] between PHY_CYA and CTRL.

In addition to these uncertainties in the prescribed light absorption strength per cyanobacteria biomass, uncertainties in the cyanobacteria distribution itself impose uncertainties on the model results. As discussed in Paulsen et al. (2017), the model underestimates cyanobacteria concentrations in the North Atlantic subtropical gyre. Related to that, the surface cooling effect in the tropical Atlantic might be underestimated. On the other hand, the model seems to overestimate cyanobacteria concentrations

in the tropical East Pacific, and thus also likely the cooling effect of the Pacific cold tongue. Despite these uncertainties in the details of the strength and patterns of the effects, the model results clearly indicate the potential of the phytoplankton group of cyanobacteria to play a relevant role in the Earth system.

## 6.2 Implications for the Earth system model

In this study, we focused on the impact of including prognostic cyanobacteria in addition to bulk phytoplankton in the dynamic

biological shortwave heating on the climate system in MPI-ESM. In the following, we discuss our results in the context of the standard model version with globally uniform optical water type (our simulation CTRL) and in the context of observations. Figure 11a shows the SST bias of CTRL in comparison to observations (the Polar Science Center Hydrographic Climatology, PHC3, a blend of the Levitus et al. (1998) data with an updated data set for the Arctic from Steele et al. (2001)). The changes in SST due to including the biogeophysical feedback from phytoplankton – bulk phytoplankton and cyanobacteria – on radiative

heating only slightly improve the globally averaged mean SST bias (0.61 K in PHY_CYA compared to 0.62 K in CTRL) and the globally averaged root mean square error (1.35 K in PHY_CYA compared to 1.38 K in CTRL). In contrast, regional improvements are much more prominent. Especially, the tropical Pacific cold bias in PHY_CYA is reduced by 0.7 K (~25 %) in comparison to CTRL (Figure 11b). This furthermore implies a reduction of the precipitation bias in the western equatorial Pacific (not shown). The reduction of the cold bias in PHY_CYA compared to CTRL can be explained as follows: The globally





constant water turbidity assumed in CTRL (which roughly refers to 0.44 mg Chl m$^{-3}$) overestimates the turbidity, and hence light absorption strength, within the clear water regions of the oligotrophic subtropical gyres. In PHY_CYA, instead, the phytoplankton-dependent water turbidity takes account of these low-chlorophyll areas. This leads to deeper reaching light penetration within the subtropical gyres which results in a warming of the subsurface water within the shallow meridional

overturning cells – the water that is eventually upwelled at the equator. This equatorial surface warming effect due to clearer subtropical gyres was also seen in the study of Wetzel et al. (2006), in which a former version of MPI-ESM was used. Also other studies (Patara et al., 2012, and references therein) compared a model state with phytoplankton light absorption included (either from satellite chlorophyll or calculated in the biogeochemical model) against a reference state with globally constant water turbidity (either corresponding to zero chlorophyll or a constant non-zero chlorophyll value). Whether this leads to

a warming, as in our study and in Wetzel et al. (2006), or to a cooling of equatorial SST, strongly depends on the chosen reference state: If the reference turbidity within the subtropical gyres is lower than the prognostic value (e.g., zero chlorophyll, as in Gnanadesikan and Anderson, 2009), including the feedback leads to a cooling. In this case the higher chlorophyll values shade and hence cool the subsurface water that is upwelled at the equator. If the reference turbidity within the subtropical gyres, on the other hand, is larger than the prognostic value (as is the case in our study and in Wetzel et al., 2006), including

the feedback leads to a warming as described in the beginning of this paragraph. The magnitudes of the anomalies are not directly comparable between the studies due to the different reference attenuation depths which were used (11–43 m, see Patara et al., 2012, and references therein). All studies, including ours, however, agree that applying a phytoplankton-dependent light attenuation scheme instead of a globally uniform attenuation has considerable effects on the simulated model state. In our case, the changes improve some of the model biases, which emphasizes the relevance of including the bio-physical feedback in the

model.

Omitting the effect of cyanobacteria (that means considering only the effect of bulk phytoplankton) results in even lower water turbidity within the subtropical gyres. The heating of the Pacific cold tongue and hence the improvement with respect to the Pacific SST bias compared to CTRL is even larger in PHY_ONLY compared to PHY_CYA. The strong regulative effect that cyanobacteria have on equatorial SST is important to note in this regard. Considering cyanobacteria in addition to bulk

phytoplankton in the feedback, changes the anomaly by ∼30 % (PHY_CYA - CTRL: 0.7 K, PHY_ONLY - CTRL: 1.0 K). This magnitude (0.3 K, Figure 2a) is roughly half of ∼0.75 K, which is the difference that arises from running the model with a higher horizontal resolution (MR, global resolution of 0.4°, Jungclaus et al., 2013). The fact that in the equatorial region more realistic SST values are simulated when omitting light absorption by cyanobacteria, might have two reasons. First, the surface chlorophyll of bulk phytoplankton in the subtropical gyres is slightly overestimated in comparison to satellite data

(as mentioned in Section 3). This might lead to an overestimation of the water turbidity and hence an underestimation of the subsurface heating and the related equatorial surface warming. Second, model parameters were chosen to best possibly represent the climate state in a model version without accounting for the biogeophysical feedback of light absorption. The fact that including cyanobacteria, which to our best knowledge should improve the represented distribution of light absorption, does, however, regionally impair the model biases, might indicate the demand for a retuning of certain model parameters, such

as the ones related to vertical mixing.



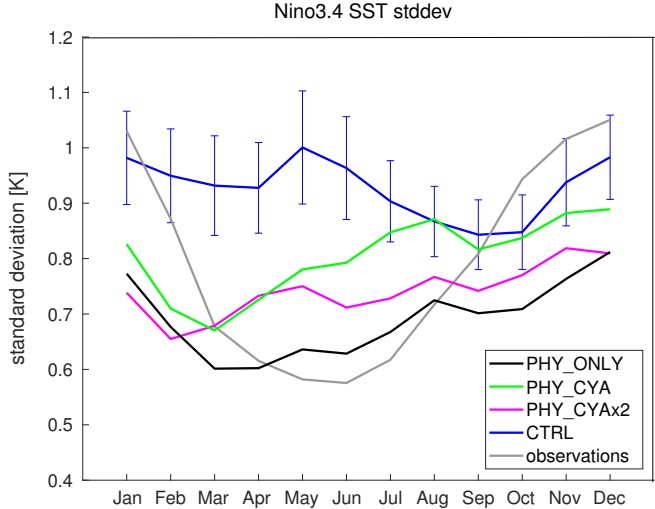

**Figure 12.** Seasonal cycle of the standard deviations of the Niño3.4 SST index (SST anomalies averaged over the area 5° S–5° N, 170° W–120° W) for the experiments PHY_ONLY (black), PHY_CYA (green), PHY_CYAx2 (puple), CTRL (blue) and observations (HadISST1) (grey). The blue errorbars on the blue curve show ± the standard deviation of eight 100 year periods of CTRL.

Associated with a reduction of the SST bias of the Pacific cold tongue in the experiments with feedback included (PHY_ONLY, PHY_CYA and PHY_CYAx2) compared to CTRL, also the representation of the seasonality of tropical Pacific variability is improved (Figure 12). Whereas in CTRL the Niño3.4 variability seems to have almost no seasonality, all simulations with feedback included establish a seasonal cycle in closer agreement with observations (HadISST1, Rayner et al., 2003). If the

differences between the setups with and without cyanobacteria are significant against the large internal variability of the model (see blue bars in Figure 12 which show ± the standard deviation of eight 100 year persiods of CTRL), has to be tested in longer runs. The improvement of the seasonality of ENSO variability due to applying a phytoplankton-dependent instead of a globally uniform light attenuation was also found in Wetzel et al. (2006).

   In contrast to the equatorial Pacific, where the inclusion of light absorption by cyanobacteria in addition to bulk phytoplank-

ton does rather impair the SST bias, there are also regions where the SST bias is improved. In the coastal upwelling region of the southeastern tropical Atlantic, the cooling effect caused by including cyanobacteria reduces the coastal warm bias by 0.2–0.4 K in PHY_CYA compared to PHY_ONLY (relative to CTRL). This magnitude, and also the bias reduction in the equatorial Pacific of 0.7 K in PHY_CYA compared to CTRL mentioned above, are significantly larger than the internal variability of the model. They do, however, not completely eliminate the model biases. Other model uncertainties, like erroneous winds

and an improper representation of stratocumulus clouds still prevail (Jungclaus et al., 2013). Furthermore, the resolution of the atmospheric (Milinski et al., 2016) and the ocean general circulation model (Jungclaus et al., 2013) might induce biases. Nevertheless, our results show that the sensitivity of SST and climate to details in the spatial distribution of light absorption by





marine biota, indeed, adds another level of complexity to the model which is largely independent of the model resolution and which should be taken into account in considerations of climate model biases.

## 7   Summary and Conclusions

We use the Earth system model MPI-ESM to study the effects of light absorption by marine cyanobacteria on the tropical climate system. We find that accounting for prognostic cyanobacteria in addition to bulk phytoplankton in the attenuation depth of light has significant effects on the model's climate mean state and variability. Cyanobacteria induce a surface cooling on tropical SST on climatological time scale in the order of 0.5 K. This is because cyanobacteria biomass, located throughout the tropical and subtropical ocean, shades and hence cools the subsurface water that is upwelled at the equator and in eastern boundary upwelling regions. In most regions, this advective process of bringing cooler subsurface water to the surface outweighs the direct local heating effect that was inferred from observations and idealized one-dimensional studies locally. The equatorial cooling leads to an expansion of the Hadley cells, a strengthening ($\sim$6 %) and westward shift ($\sim$3.0° longitude) of the Walker circulation, as well as changes in the precipitation patterns by up to 1.0 mm d$^{-1}$. Changes in ocean temperature and circulation feed back on cyanobacteria growth itself, imposing complex patterns of positive and negative feedback loops. Furthermore, the amplitude of the seasonal cycle of SST in areas where cyanobacteria are abundant is increased by $\sim$25 %. Tropical Pacific variability is enhanced by roughly 20 %. The magnitudes of all effects are thereby sensitive to the prescribed strength of light absorption per cyanobacteria biomass. Doubling the chlorophyll content per cyanobacteria biomass (which is a value in the upper range of observations), roughly doubles the magnitudes of most effects.

Accounting for the biologically induced radiative heating of phytoplankton (bulk phytoplankton and cyanobacteria) improves some of the model SST biases, e.g., the Pacific cold bias and the coastal warm bias in the southeastern tropical Atlantic, compared to the standard model version which applies a globally uniform light attenuation depth. The effects between including or not including cyanobacteria in the feedback demonstrate the sensitivity of the simulated climate mean state and variability to details in the distribution and representation of marine biota. In light of the uncertainties in the spatial distribution and light absorption strength of phytoplankton, the results stress the need for more observations to further constrain and better represent the effects of light absorption by marine biota in Earth system models. Our results indicate that the functional group of surface buoyant, N$_2$-fixing cyanobacteria is important to be considered due to its regulative effect on tropical SST and climate.

*Code and data availability.*   All simulations were performed at the German Climate Computing Center (DKRZ). The model code, primary data and scripts needed to reproduce the analyses presented in this study are archived by the Max Planck Institute for Meteorology, and will be available by contacting publications@ mpimet.mpg.de.

*Author contributions.*   H.P. developed the model code (with contributions from T.I., K.D.S. and I.S.), performed the experiments, and carried out the analyses (with contributions from all co-authors). H.P. prepared the manuscript with contributions from all co-authors.

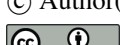

*Competing interests.* The authors declare that they have no conflict of interest.

*Acknowledgements.* This study was part of the PhD thesis of H.P. (Paulsen, 2018) and was funded by the International Max Planck Research School on Earth System Modelling. The authors would like to thank Michael Botzet for performing the simulation CTRL and Sebastian Sonntag for helpful comments on the manuscript.



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
