# Peer review of "Light absorption by marine cyanobacteria affects tropical climate mean state and variability"

_Earth System Dynamics, 2018_

## Referee Comment (RC1) · Anonymous Referee #1 · 11 Oct 2018

Summary: This manuscript presents a sensitivity study of the effects of including a biophysical feedback of general phytoplankton and cyanobacteria light attenuation in an earth system model. With respect to cyanobacteria, the authors find a significant effect on tropical general circulation that opposes the prevailing understanding based on observations and regional modelling. Biophysical feedbacks are typically ignored in ocean modelling, thus this manuscript advances our understanding of potential deficits in the state of the science. The science is of a high quality and is suitable for publication in ESD. The language needs editing but I have no major concerns about this study.

General Comments:

The positive buoyancy of cyanobacteria in the model is novel, as far as I am aware, and deserves more discussion. If a high concentration of cyanobacteria warms the

local water (or produces a cooling via advection), this should also change the surface density. I assume the cyanobacteria buoyancy then adjusts, but are there upper or lower limits to this? If limits are imposed, is there an implication for the model results?

Specific Comments:

Abstract: Please add a sentence explaining the reason for the apparent disagreement between your model results (cooling effect) and observations (heating effect).

Page 2, line 14-16. Not quite true, see Dutkiewicz et al. (2015) Biogeosciences and references contained within. I'm not sure how many of these include bio-physical feed-back however.

Page 18 line 16. The model has short time steps. Does it produce temporary local heating? And please mention whether there has been observed subsurface cooling associated with the localized warming.

Technical Corrections:

Abstract line 4: remove "as of yet".

Abstract line 5: Please specify whether these previous studies are model studies or observation-based.

Abstract lines 10-11: This sentence is awkward. I recommend removing "the specific phytoplankton group of" to make the subject more clear.

Abstract line 12: remove "the" from before "phytoplankton-dependent"

Page 2 line 2-3: remove either "over" or "up to", and change to: "several million square kilometres"

Page 2 line 13: remove "indeed"

Page 2 line 14: "at the best" should be "or at best"

Page 2 line 19: edit to "to particularly affect", and insert the relevant regions associated

with each study for clarity.

Page 2 line 22: insert a comma after "circulation"

Page 2 line 23: remove "identified to be relevant"

Page 2 line 24: insert "also" before "play" and remove extra "e" in "globale"

Page 2 lines 27-28: insert commas after "cyanobacteria" and "phytoplankton"

Page 2 line 29: insert comma after "phytoplankton" and "the" after "address"

Page 2 lines 29-33: Please add numbers to the questions to separate them more clearly. The first 2 questions should be made more specific. Question 4: "What are the positive/negative..."

Page 3 line 18: So there is a single shortwave radiation value at each latitude per day? This should be made clearer.

Page 3 line 22: remove "which"

Page 3 line 23: insert "and" before "includes"

Page 3 line 24: remove "the compartments" and parentheses

Page 3 line 25: add "two" and remove the contents of the parentheses- this is repeated in the next paragraph.

Page 3 line 31-32. This sentence is repetitive and not needed.

Page 3 line 32: remove "and, on the other hand," and replace with "but can also". The next sentence is also repetitive and not needed.

Page 4 line 1: What other models include positive buoyancy in some phytoplankton types? This should be addressed by one sentence in the Introduction.

Page 4 Equation 1: There is no light attenuation by sea ice? Are the parameter values also adopted from Zielinski et al (2002) or were they tuned for MPI-ESM according to

some criteria?

Page 5 Table 1 and line 12: replace "set to" in PHY_ONLY with "made"

Page 6 line 1: "contents" should be singular

Page 6 line 11: remove "anymore"

Page 6 line 12: commas after "cyanobacteria" and "phytoplankton"

Page 6 line 14: "consulted" is the wrong word. Perhaps "... years and represents an estimate of the internal..."

Page 6 line 20: "optimum temperature" for what?

Page 6 line 25: replace "thereby" with "therefore"

Page 6 line 26: edit to "The order of magnitude of both biomass and N2 fixation rates, as well as the large scale spatiotemporal patterns, ..."

Page 6 line 29: remove "probably somewhat"

Page 7 Figure 1 caption: spelling error in "physical"

Page 7 line 11: "set to" should be "made"

Page 8 line 7: really? My impression from Page 6 line 25 and Page 7 is the top layer is 22 m thick.

Page 8 line 9: "refer" is the wrong word- "convert" instead?

Page 8 line 11: replace "in" with "of"

Page 8 line 14: reverse order of "reach now" and delete everything after 1.6 mg Chl m-3.

Page 8 line 20: Change sentence to: "In contrast, a cooling effect on SST dominates larger areas."

Page 8 line 22: insert comma after "PHY_CYA"

Page 10 line 1: replace "an" with "the"

Page 10 line 6: insert "physical" before "regimes"

Page 10 line 9: I think you mean "west of" not "western"

Page 10 line 13: remove "and" after "gyres"

Page 10 line 15: edit to "...dominates large parts..."

Page 10 line 16: remove "as mentioned above"

Page 11 line 4: edit to "Here it is not the local effect of high light absorption causing the anomalies, but changes..."

Page 11 line 6: remove comma after "experiment"

Page 11 line 8: edit to "are, enlarged by roughly..."

Page 11 line 11: remove second "which"

Page 11 line 13: change "which" to "that"

Page 13 lines 30 and 32 and Page 15: change "dampens" to "damps"

Page 13 line 32: edit to "by causing less transport of cold subsurface..."

Page 14 lines 4-5: edit to "... strength, but damps it in the ..."

Page 15 line 15: remove "rather"

Page 15 line 17: replace "different" with "variable"

Page 16 line 7: remove "exemplarily"

Page 17 line 4: move "also" to before "affected"

Page 18 line 5 and line 13: "let expect" is awkward. One solution: "Interestingly, a

stronger cooling ... PHY_CYA does not lead to a stronger..." And then remove "however" from the next sentence. Line 13: remove "would let expect" and replace with "suggest" and change "to" to "would"

Page 18 line 7 and 12: "which" should be "that"

Page 18 line 16: "episodic"

Page 18 line 18: "larger", "local surface heating effect", remove "acting"

Page 19 line 8: remove "especially the" and replace with "explicitly accounting for"

Page 19 line 11: "Agreement on the sensitivity"..."circulation, as well as precipitation patterns,"...

Page 19 line 17: insert "and" before 3)

Page 19 line 20: replace "which" with "that"

Page 19 line 25: replace "which" with "that" and "underpin with numbers" with "quantify"

Page 20 line 1: "retroaction" is a strange word. Perhaps "effect of light attenuation"?

Page 20 line 6: "which" should be "that"

Page 20 line 18: remove "rather"

Page 20 line 19: "diverging"

Page 20 line 22: "linearly"

Page 21 line 4: remove "seems to", change to "overestimates"

Page 21 line 17: remove "Especially,"

Page 21 line 19: third "the" should not be capitalized

Page 22 line 1: replace "roughly refers" with "approximates"

Page 22 line 4: "..that results.."

Page 22 line 6: move "also" to before "compared"

Page 22 line 15: remove "as described in the beginning of this paragraph"

Page 22 line 31: remove "possibly"

Page 23 line 2: move "also" to before "improved"

Page 24 line 10: remove "locally"

Page 24 line 20: change "which" to "that"

---

## Referee Comment (RC2) · Anonymous Referee #2 · 14 Oct 2018

Paulsen et al. implement light absorption by cyanobacteria in the MPI ESM and discuss its impact on the mean climate as well as on the seasonal SST cycle and interannual SST variability. They suggest that cyanobacteria lower SST due to a shading effect: the sub-surface water cool because they receive less sunlight, and the upwelling of these sub-surface waters lead to a SST decrease. Changes in SST gradients induce changes in ocean currents, thus slightly altering the mean climate state. Light absorption by cyanobacteria also leads to a greater seasonal SST cycle in the tropics.

Paulsen et al., implement a new effect, that is neglected in most models I believe. The parametrization implemented makes sense and the MPI model is well tested. The impact on sub-surface temperature seem justified. The manuscript is clear and well written. I thus recommend publication after the comments below are addressed.

[Figure]

1) This new implementation affects the oceanic and atmospheric circulations with effects reaching the mid latitude. Explaining some of these changes is not straight forward, and the authors suggest changes in the Hadley cells and the Walker circulation. In the present state of the manuscript it is not possible to check whether this is really the case. Changes in the Hadley cells and the Walker circulation would need to be shown: figures of the atmospheric circulation are needed. Figures showing changes in ocean surface currents would also help as they are discussed in the text.

If the Walker circulation is stronger (p11, L.17), shouldn't the upwelling in the Eastern equatorial pacific be stronger (p10, L.18)? From Fig. 6, I can see that the upwelling in the EEP and EEA are indeed weaker, and that the barotropic stream-function seems weaker almost everywhere. This is consistent with weaker wind-driven circulation as mentioned p13, l12. I can see that in other parts of the text the authors suggest the strengthening is mostly restricted to the western side of the Pacific basin. Please make sure your description of the Walker circulation changes are accurate. The northward shift of the Gulf Stream is consistent with Fig.6b, however the change in the Kuroshio is less clear.

P13, L.1: It is stated that the Hadley cells are expanded: is it in both hemispheres?

P13, L. 19-20: This sentence is really confusing. Are the authors really talking about the Northwest Pacific Ocean? Is there really southerly winds dominating there (I would have thought it would have been westerlies)? Latitude and an East-West location are needed to really understand this sentence. As mentioned above a figure of winds and wind anomalies would really help to understand.

2) Section 5: seasonal dynamics. It might help to mention at which time the cyanobacteria blooms occur. The timing of the blooms will most likely significantly impact the seasonal changes. Is the timing of the blooms in agreement with observations?

3) Figures: Figure 1: Add Latitude in plot 1b as it is different from a, c and d It would be nice to add the satellite estimate of Chla for comparison.

Figure 2: why are the plots cut at 50deg? It might help to see what's happening poleward of 50deg, particularly for the change in Kuroshio and Gulf Stream.

Figure 11: Should the authors also plot PHY_ONLY-CTRL? In that way in one figure one can see the impact varying light attenuation, and the impact of the cyanobacteria. This would be particularly useful as the high latitude regions are not shown in Figure 2.

4) Minor and typos: P6, L. 13: "Furthermore, we…" P6, L. 33: "For comparison…" (remove the) P10, L.19: "…surface warming." (remove effect) P11, L. 4-5: Rephrase "Here, it is probably the changes in the circulation system that is causing the anomalies instead of the local heat absorption effect." P13, L.3: "There" instead of "Here" P13, L. 21: "Interior"

Section 6: Many sentences have a weird structure. It would be worth trying to improve the flow of the section.
* * *

---

## Author Comment (AC1) · 21 Nov 2018

**Light absorption by marine cyanobacteria affects tropical climate mean state and variability**

**Reply to Reviewer 1**

Hanna Paulsen and Co-authors

November 2018

We thank the reviewer for the positive feedback and for the helpful and constructive comments. A point-to-point reply is given below. The referee's comments are in red color, and our reply in black color.

Summary: This manuscript presents a sensitivity study of the effects of including a biophysical feedback of general phytoplankton and cyanobacteria light attenuation in an earth system model. With respect to cyanobacteria, the authors find a significant effect on tropical general circulation that opposes the prevailing understanding based on observations and regional modelling. Biophysical feedbacks are typically ignored in ocean modelling, thus this manuscript advances our understanding of potential deficits in the state of the science. The science is of a high quality and is suitable for publication in ESD. The language needs editing but I have no major concerns about this study.

General Comments: The positive buoyancy of cyanobacteria in the model is novel, as far as I am aware, and deserves more discussion. If a high concentration of cyanobacteria warms the local water (or produces a cooling via advection), this should also change the surface density. I assume the cyanobacteria buoyancy then adjusts, but are there upper or lower limits to this? If limits are imposed, is there an implication for the model results?

In the model, we assume a constant rising velocity of 1 m d$^{-1}$. Thus, density changes of the surrounding water do not feed back on the buoyancy of cyanobacteria. The vertical movement of cyanobacteria is not well constrained, and observations range from a negative buoyancy (sinking) to a rising velocity of about 260 m d$^{-1}$ (Walsby, 1978; Villareal and Carpenter, 2003; Guidi et al., 2012). The aim of prescribing a positive rising velocity (instead of a neutral buoyancy as prescribed for bulk phytoplankton) is to simulate accumulations of cyanobacteria biomass at the surface (representing surface blooms). In Paulsen et al. (2017), we did sensitivity experiments with varying rising velocities (ranging from 0 to 50 m d$^{-1}$). We found the largest sensitivity between 0 m d$^{-1}$ and 1 m d$^{-1}$. Already a value of 1 m d$^{-1}$ counteracts the downward mixing and leads to a major fraction of biomass in the first model layer. As higher rising velocities give only slightly different results, we decided to take a value of 1 m d$^{-1}$, analogous to Sonntag (2013).

Specific Comments:

Abstract: Please add a sentence explaining the reason for the apparent disagreement

between your model results (cooling effect) and observations (heating effect).

Observations rather represent snapshots of local heating events. In the model, we also find a local increase in SST on short timescales in times of high cyanobacteria concentrations. On larger timescales, however, the cooling due to the upwelling of cooler subsurface water outweighs the local warming effect in largest areas. The focus of this paper is on these global effects on climatological timescales which leads to the apparent disagreement between the observed local heating and the simulated cooling. We slightly modified the sentences in the abstract, so that the difference between observations (short time scales) and presented model results (climatological timescales) is made more clear.

Page 2, line 14-16. Not quite true, see Dutkiewicz et al. (2015) Biogeosciences and references contained within. I'm not sure how many of these include bio-physical feedback however.

The study of Dutkiewicz et al. (2015) explicitly includes the effect of several optically important water constituents (amongst others different phytoplankton functional types) to resolve the penetration of spectral irradiance. The study, however, does not include the feedback from phytoplankton light absorption on temperature, but only investigates the effects of different light attenuation formulations on ocean biogeochemistry. In the respective text passage in our manuscript we emphasize that to our knowledge there is no study that differentiates between different phytoplankton types and investigates their contribution to the feedback on physics. We modified the text in the manuscript to make this more clear to the reader (page 2 lines 14-16).

Page 18 line 16. The model has short time steps. Does it produce temporary local heating? And please mention whether there has been observed subsurface cooling associated with the localized warming.

The increased light absorption due to the presence of cyanobacteria always leads to a relative local heating of the surface water. If the net effect is, however, a warming or a cooling depends on which of the two processes – the warming or the cooling due to upwelling and transport of cooler subsurface water – dominates. On short timescales, we indeed see the local heating effect which takes place in times of high cyanobacteria concentrations. On climatological timescale, the timescale we focus on in this study, the surface cooling prevails in largest areas. There are, however, some limited regions, such as the eastern tropical Atlantic and Pacific, with a positive SST anomaly on the climatological mean (as described in the manuscript, page 10, lines 16-20).

The observations focused on the surface ocean (Kahru et al., 1993; Capone et al., 1998; Wurl et al., 2018). The respective studies only reported a surface heating effect and do not mention the effect on the subsurface. It is probably difficult to attribute measured subsurface anomalies to the impact of cyanobacteria light absorption due to other processes acting at the same time.

Technical Corrections:

We thank the reviewer for the detailed corrections of the language. We changed everything as recommended. In the following, we only list the points that require additional commenting by the authors.

Page 2 line 19: edit to "to particularly affect", and insert the relevant regions associated with each study for clarity.

We edited the formulation (page 2 lines 18-19). All mentioned studies, however, focus on similar regions. So, it is not necessary to separate between the studies.

Page 3 line 18: So there is a single shortwave radiation value at each latitude per day? This should be made clearer.

As we are using a fully coupled Earth system model, incoming shortwave radiation that enters the ocean depends on regionally varying factors, such as cloud conditions. It is, however, correct that the atmospheric model is coupled to the ocean model once per day, that means there is no diurnal cycle in the incoming radiation. We added this in the manuscript (page 4 line 10).

Page 4 line 1: What other models include positive buoyancy in some phytoplankton types? This should be addressed by one sentence in the Introduction.

The studies which are given as references in the manuscript (Hense, 2007; Sonntag and Hense, 2011; Sonntag, 2013), use an idealized biogeochemical model which includes positive buoyancy of cyanobacteria. It is applied in an one-dimensional as well as three dimensional setup of the North Atlantic, and the Baltic Sea, respectively. We already mention in the Introduction that these studies include the positive buoyancy of cyanobacteria (page 2 line 9). To our knowledge there is no global model that includes positive buoyancy of phytoplankton types. We modified the respective text passage in the Introduction to make clear, that we refer to global model studies in this context (page 2 line 16).

Page 4 Equation 1: There is no light attenuation by sea ice? Are the parameter values also adopted from Zielinski et al (2002) or were they tuned for MPI-ESM according to some criteria?

Light is not able to penetrate through sea ice in the model. Equation 1 refers to ice-free regions of the ocean.

Yes, the parameter values for light attenuation are adopted from Zielinski et al. (2002). We added one sentence in the manuscript (page 4 lines 15-16).

Page 6 line 29: remove "probably somewhat"

In the eastern tropical Pacific, there are no observations of cyanobacteria biomass and $N_2$ fixation available. The statement that the model overestimates biomass and $N_2$ fixation rates is only based on the fact, that the dust deposition is overestimated in this region (Paulsen et al., 2017). We reformulated the sentence to make this more clear (page 6 lines 27-29): "In the eastern tropical Pacific, on the other hand, concentrations and fixation rates are high, but the lack of observational data does not allow for a proper assessment."

Page 8 line 7: really? My impression from Page 6 line 25 and Page 7 is the top layer is 22 m thick.

The first model layer is 12 m thick, modified by variations of the sea surface height. The figures in the manuscript show the mean of the first two layers (∼22 m). The caption of Figure 1 of the manuscript, which said "surface concentrations", was indeed somewhat misleading. We modified it accordingly to "in the upper 22 m" to make it clear (see caption of Figure 1 in the manuscript, page 7).

Page 21 line 4: remove "seems to", change to "overestimates"

As mentioned above, in the eastern tropical Pacific there are not sufficient observations to evaluate the model performance with respect to cyanobacteria concentrations. We therefore reformulate it to "might overestimate" (page 20 line 34).

**References**

Capone, D. G., A. Subramaniam, J. P. Montoya, M. Voss, C. Humborg, A. M. Johansen, R. L. Siefert, and E. J. Carpenter (1998). An extensive bloom of the diazotrophic cyanobacterium, *Trichodesmium*, in the Central Arabian Sea during the spring inter-monsoon. *Mar. Ecol. Prog. Ser 172*, 281–292.

Dutkiewicz, S., A. Hickman, O. Jahn, W. Gregg, C. Mouw, and M. Follows (2015). Capturing optically important constituents and properties in a marine biogeochemical and ecosystem model. *Biogeosciences 12*(14), 4447–4481.

Guidi, L., P. H. Calil, S. Duhamel, K. M. Björkman, S. C. Doney, G. A. Jackson, B. Li, M. J. Church, S. Tozzi, Z. S. Kolber, et al. (2012). Does eddy-eddy interaction control surface phytoplankton distribution and carbon export in the North Pacific Subtropical Gyre? *Journal of Geophysical Research: Biogeosciences 117*(G2).

Hense, I. (2007). Regulative feedback mechanisms in cyanobacteria-driven systems: a model study. *Marine Ecology Progress Series 339*, 41–47.

Kahru, M., J.-M. Lepppaenen, and O. Rud (1993). Cyanobacterial blooms heating of the sea surface. *Marine ecology progress series. Oldendorf 101*(1), 1–7.

Paulsen, H., T. Ilyina, K. D. Six, and I. Stemmler (2017). Incorporating a prognostic representation of marine nitrogen fixers into the global ocean biogeochemical model HAMOCC. *Journal of Advances in Modeling Earth Systems 9*(1), 438–464.

Sonntag, S. (2013). *Modeling biological-physical feedback mechanisms in marine systems*. Ph. D. thesis, Universität Hamburg.

Sonntag, S. and I. Hense (2011). Phytoplankton behavior affects ocean mixed layer dynamics through biological-physical feedback mechanisms. *Geophysical Research Letters 38*(15).

Villareal, T. and E. Carpenter (2003). Buoyancy regulation and the potential for vertical migration in the oceanic cyanobacterium *Trichodesmium*. *Microbial Ecology 45*(1), 1–10.

Walsby, A. (1978). The properties and buoyancy-providing role of gas vacuoles in *Trichodesmium* Ehrenberg. *British Phycological Journal 13*(2), 103–116.

Wurl, O., K. Bird, M. Cunliffe, W. Landing, U. Miller, N. Mustaffa, M. Ribas-Ribas, C. Witte, and C. Zappa (2018). Warming and inhibition of salinization at the ocean's surface by cyanobacteria. *Geophysical Research Letters*.

Zielinski, O., O. Llinás, A. Oschlies, and R. Reuter (2002). Underwater light field and its effect on a one-dimensional ecosystem model at station ESTOC, north of the Canary Islands. *Deep Sea Research Part II: Topical Studies in Oceanography 49*(17), 3529–3542.

---

## Author Comment (AC2) · 21 Nov 2018

**Light absorption by marine cyanobacteria affects tropical climate mean state and variability**

**Reply to Reviewer 2**

Hanna Paulsen and Co-authors

November 2018

We thank the reviewer for the positive feedback and for the helpful and constructive comments. A point-to-point reply is given below. The referee's comments are in red color, and our reply in black color.

Paulsen et al. implement light absorption by cyanobacteria in the MPI ESM and discuss its impact on the mean climate as well as on the seasonal SST cycle and interannual SST variability. They suggest that cyanobacteria lower SST due to a shading effect: the sub-surface water cool because they receive less sunlight, and the upwelling of these sub-surface waters lead to a SST decrease. Changes in SST gradients induce changes in ocean currents, thus slightly altering the mean climate state. Light absorption by cyanobacteria also leads to a greater seasonal SST cycle in the tropics.

Paulsen et al., implement a new effect, that is neglected in most models I believe. The parametrization implemented makes sense and the MPI model is well tested. The impact on sub-surface temperature seem justified. The manuscript is clear and well written. I thus recommend publication after the comments below are addressed.

1) This new implementation affects the oceanic and atmospheric circulations with effects reaching the mid latitude. Explaining some of these changes is not straight forward, and the authors suggest changes in the Hadley cells and the Walker circulation. In the present state of the manuscript it is not possible to check whether this is really the case. Changes in the Hadley cells and the Walker circulation would need to be shown: figures of the atmospheric circulation are needed. Figures showing changes in ocean surface currents would also help as they are discussed in the text.

Figures of the Hadley circulation (zonal global mean meridional mass flux), surface wind, wind stress on the ocean, and ocean surface currents – as well as their respective anomalies – are provided at the end of this document (Figures R 1-4). They are discussed in more detail below.

If the Walker circulation is stronger (p11, L.17), shouldn't the upwelling in the Eastern equatorial pacific be stronger (p10, L.18)? From Fig. 6, I can see that the upwelling in the EEP and EEA are indeed weaker, and that the barotropic stream-function seems weaker almost everywhere. This is consistent with weaker wind-driven circulation as mentioned p13, l12. I can see that in other parts of the text the authors suggest the

strengthening is mostly restricted to the western side of the Pacific basin. Please make sure your description of the Walker circulation changes are accurate. The northward shift of the Gulf Stream is consistent with Fig.6b, however the change in the Kuroshio is less clear.

As stated correctly by the reviewer, the strengthening of the Walker circulation is restricted to the western Pacific (visible in the strengthening of the westward equatorial winds, Figure R 2). At the same time, the transition between convection and subsidence is shifted towards the west (see Figure 5c of the manuscript). That means, the region of convection is more confined to the western part of the Pacific basin. Related to that, the strengthening of the windstress on the ocean (Figure R 3) and the strengthening of the equatorial upwelling (Figure 6d of the manuscript) is also restricted to the western equatorial region. In the eastern tropical Pacific, on the other hand, the weakening of the trade winds (Figure R 2) due to the weaker meridional SST gradient rather causes a decrease in upwelling strength (Figure 6d of the manuscript). We modified the respective text passages in the manuscript to be more precise about the changes in the Walker circulation (page 11 lines 14-20, page 13 lines 1-2). Furthermore, we included figures of the wind field and its anomaly (Figure R 2) in the manuscript (Figure 5a,b).

The northward shift of the subtropical gyre (and the western boundary current) is indeed more pronounced in the North Atlantic than in the North Pacific, as seen in the barotropic streamfunction (Figure 6b of the manuscript). Although a slight northward shift is also visible in the North Pacific (better visible in experiment PHY_CYAx2 than in PHY_CYA, see Figure R 5), the dominant cause of the cold anomaly in the western subtropical Pacific is probably the decrease in the northward transport of warm tropical surface water (Figure R 4) related to the changes in the wind field (Figure R 2). We modified the respective passage in the manuscript to make this more clear (page 12 lines 32-33, page 14 lines 1-2 ).

Although the additional figures (Figures R 1-4 of this document) generally underline what is described in the text, we decided to add only one additional figure to the manuscript, showing the wind field and its anomaly (Figure R 2 of this document, Figure 5a,b in the manuscript, respectively). The other properties (wind stress on the ocean and ocean surface currents) mainly follow the patterns of the surface wind field and hence do not need to be additionally shown in our opinion. Following the reviewer comments, we improved the descriptions in the text regarding the changes in Hadley and Walker circulation.

P13, L.1: It is stated that the Hadley cells are expanded: is it in both hemispheres?
Figure R 1 shows the global zonal mean meridional mass flux $\Psi$ visualizing the Hadley circulation. In both hemispheres, the anomaly patterns indicate a slight shift of the boundaries of the Hadley cells polewards. The expansion is, however, rather small. Furthermore, in the southern hemisphere, the subtropical gyres do not show a poleward shift in the barotropic streamfunction (Figure 6b of the manuscript) which one would expect from an expansion of the Hadley cells. We modified the respective text passages in the manuscript and stress the weakening of the Hadley circulation rather than the expansion of its cells (page 1 line 8, page 13 line 6, page 24 line 5).

P13, L. 19-20: This sentence is really confusing. Are the authors really talking about

the Northwest Pacific Ocean? Is there really southerly winds dominating there (I would have thought it would have been westerlies)? Latitude and an East-West location are needed to really understand this sentence. As mentioned above a figure of winds and wind anomalies would really help to understand.

Winds and wind anomalies are shown in Figure R 2 and are added to the manuscript (Figure 5a,b). The description in the manuscript was indeed not very clear about the geographical location of the described changes. The location that is meant is in the tropical Pacific north of the equator along the coast of Southeast Asia (10-30°N, ∼120°E). In this region, a southwestward wind is prevailing on the climatological mean (Figure R 2a). In PHY_CYA and PHY_CYAx2, this wind is enhanced (Figure R 2b,c), subsequently enhancing the windstress on the ocean (Figure R 3b,c). This southwestward windstress acts in the opposite direction than the flow of the northeastward western boundary current (Figure R 4a) and hence reduces its transport. This, together with the slight northward shift of the northern boundary of the subtropical gyre (Figure 6b of the manuscript and Figure R 5 of this document), results in the cold SST anomaly in the northwestern Pacific subtropical gyre. We modified the respective text passage in the manuscript to make the location of the described changes of winds and ocean transport more clear (page 13 lines 26-29).

2) Section 5: seasonal dynamics. It might help to mention at which time the cyanobacteria blooms occur. The timing of the blooms will most likely significantly impact the seasonal changes. Is the timing of the blooms in agreement with observations?

As shown in Figure 9 of the manuscript, the timing of the blooms affects the seasonal cycle of SST in the model. The seasonal cycle of cyanobacteria concentrations is regionally quite different and depends on several factors such as temperature, ocean circulation, and phosphate and iron availability. Not many long-term observations of the seasonality of cyanobacteria concentrations and $N_2$ fixation rates are available. Exceptions regarding $N_2$ fixation are the stations ALOHA and BATS. An evaluation of the timing for ALOHA and BATS for $N_2$ fixation is given in Paulsen et al., 2017. The modeled seasonality of $N_2$ fixation roughly reproduces the observed seasonal cycle. We added some more information on the seasonality of $N_2$ fixation in the manuscript (page 6 line 29-30).

3) Figures: Figure 1: Add Latitude in plot 1b as it is different from a, c and d. It would be nice to add the satellite estimate of Chla for comparison.

We thank the reviewer for pointing this out. We modified the latitudinal range in Figure 1b so that it is identical with a,c,and d.

In this study, we specifically investigate the sensitivity of the climate system to light absorption by cyanobacteria. Satellite products of chlorophyll only provide an approximation of total chlorophyll concentrations. There is no satellite product available providing separate chlorophyll data for cyanobacteria. We, instead, evaluate the simulated distribution of cyanobacteria via its biomass (see Paulsen et al., 2017). Chlorophyll is only used as a measure of strength of light absorption. It is not a prognostic variable but depends linearly on cyanobacteria concentrations. The applied C:Chl ratio is based on observations. A sensitivity experiment was performed to assess the sensitivity to this parameter. We think that the distribution of cyanobacteria and their chlorophyll concentrations are evaluated to the extent possible. The model's ability to simulate large scale patterns of phytoplankton in general has been shown elsewhere (e.g., Wetzel et al., 2006). We therefore refrain from including a figure of satellite-derived chorophyll *a* in this study.

Figure 2: why are the plots cut at 50deg? It might help to see what's happening poleward of 50deg, particularly for the change in Kuroshio and Gulf Stream.

Figure R 6 of this document shows Figure 2 of the manuscript, but with extended latitudinal range up to 90°S and 90°N. There are indeed also anomalies in higher latitudes. However, since this paper focuses on the effects on low and mid-latitudes, we decided to keep the figures in the manuscript unchanged.

Extending the latitudinal range does not help a lot to explain the changes in the Kuroshio Current and the Gulf Stream. Thus, instead of showing a larger latitudinal range, we improved the descriptions in the text to make the changes – and its causes – more clear.

Figure 11: Should the authors also plot PHY_ONLY-CTRL? In that way in one figure one can see the impact varying light attenuation, and the impact of the cyanobacteria. This would be particularly useful as the high latitude regions are not shown in Figure 2.

Figure R 7 shows the difference in SST between PHY_ONLY and CTRL. The differences between this plot and Figure 11b of the manuscript (showing the difference between PHY_CYA and CTRL) are difficult to see since the anomalies between the experiments with and without feedback included (PHY_ONLY and CTRL, and PHY_CYA and CTRL, respectively) are larger than the anomalies between the experiments with and without cyanobacteria (PHY_ONLY and PHY_CYA). The applied color scale is hence not suitable to show both impacts – the impact of varying light attenuation, and the impact of the cyanobacteria. Thus, we decided not to include this additional figure in the manuscript.

4) Minor and typos: P6, L. 13: "Furthermore, we" P6, L. 33: "For comparison" (remove the) P10, L.19: "surface warming." (remove effect) P11, L. 4-5: Rephrase "Here, it is probably the changes in the circulation system that is causing the anomalies instead of the local heat absorption effect." P13, L.3: "There" instead of "Here" P13, L. 21: "Interior"

We modified all of the respective text passages according to the suggestions.

Section 6: Many sentences have a weird structure. It would be worth trying to improve the flow of the section

We reworked the structure of a number of sentences of Section 6 to improve their readability (pages 18-23).

[Figure]

[Figure]

[Figure]

**Figure R 1:** a) Climatological global zonal mean meridional mass flux $\Psi$ $[10^{10}$ kg s$^{-1}]$ in PHY_ONLY. The zero-isoline is overlaid in black. b) The difference in the climatological zonal mean meridional mass flux $[10^{10}$ kg s$^{-1}]$ between PHY_CYA and PHY_ONLY. The zero isoline of PHY_ONLY is overlaid in black. c) The difference in the climatological zonal mean meridional mass flux $[10^{10}$ kg s$^{-1}]$ between PHY_CYAx2 and PHY_ONLY. The zero isoline of PHY_ONLY is overlaid in black.

[Figure]

[Figure]

[Figure]

**Figure R 2:** a) Vectors: Climatological annual mean wind vectors in PHY_ONLY (the reference vector refers to 8 m s$^{-1}$). Colors: Climatological annual mean surface air temperature [K]. b) analogous to a) but for the anomalies between PHY_CYA and PHY_ONLY (the reference vector refers to 0.4 m s$^{-1}$). c) analogous to a) but for the anomalies between PHY_CYAx2 and PHY_ONLY (the reference vector refers to 0.4 m s$^{-1}$).

[Figure]

**Figure R 3:** a) Vectors: Climatological annual mean windstress on the ocean in PHY_ONLY (the reference vector refers to 0.145 N m$^{-2}$). Colors: Climatological annual mean SST [K]. b) analogous to a) but for the anomalies between PHY_CYA and PHY_ONLY (the reference vector refers to 0.005 N m$^{-2}$). c) analogous to a) but for the anomalies between PHY_CYAx2 and PHY_ONLY (the reference vector refers to 0.005 N m$^{-2}$).

[Figure]

**Figure R 4:** a) Vectors: Climatological annual mean ocean surface currents in PHY_ONLY (the reference vector refers to 0.275 m s$^{-1}$). Colors: Climatological annual mean SST [K]. b) analogous to a) but for the anomalies between PHY_CYA and PHY_ONLY (the reference vector refers to 0.02 m s$^{-1}$). c) analogous to a) but for the anomalies between PHY_CYAx2 and PHY_ONLY (the reference vector refers to 0.02 m s$^{-1}$).

[Figure]

**Figure R 5:** Difference in the barotropic streamfunction Ψ [Sv] between PHY_CYAx2 and PHY_ONLY.

[Figure]

**Figure R 6:** a) Difference in the climatological annual mean SST [K] between PHY_CYA and PHY_ONLY and b) between PHY_CYAx2 and PHY_ONLY. c) Difference in the climatological annual mean temperature at a depth of 100 m [K], and d) mixed layer depth [m] between PHY_CYA and PHY_ONLY. Dotted areas show anomalies larger than 90 % significance (Student's *t*-test).

[Figure]

**Figure R 7:** Difference in SST [K] between PHY_ONLY and CTRL.

**References**

Paulsen, H., T. Ilyina, K. D. Six, and I. Stemmler (2017). Incorporating a prognostic representation of marine nitrogen fixers into the global ocean biogeochemical model HAMOCC. *Journal of Advances in Modeling Earth Systems 9*(1), 438–464.

Wetzel, P., E. Maier-Reimer, M. Botzet, J. Jungclaus, N. Keenlyside, and M. Latif (2006). Effects of ocean biology on the penetrative radiation in a coupled climate model. *Journal of Climate 19*(16), 3973–3987.

---

## Author Comment (AC4) · 21 Nov 2018

Please see the attached PDF.

Please also note the supplement to this comment:
https://www.earth-syst-dynam-discuss.net/esd-2018-65/esd-2018-65-AC4-supplement.pdf